# Sharp Convergence Analysis of Gradient Descent for Overparameterized Deep Linear Neural Networks

## Abstract

This paper presents sharp rates of convergence of the gradient descent (GD) method for overparameterized deep linear neural networks with different random initializations. This study touches upon one major open theoretical problem in machine learning–why deep neural networks trained with GD methods are efficient in many practical applications? While the solution of this problem is still beyond the reach of general nonlinear deep neural networks, extensive efforts have been invested in studying relevant questions for deep linear neural networks, and many interesting results have been reported to date. For example, recent results on loss landscape show that even though the loss function of deep linear neural networks is non-convex, every local minimizer is also a global minimizer. When the GD method is applied to train deep linear networks, it's convergence behavior depends on the initialization. In this study, we obtained sharp rate of convergence of GD for deep linear networks and demonstrated that this rate does not depend on the types of random initialization. Furthermore, here, we show that the depth of the network does not affect the optimal rate of convergence, if the width of each hidden layer is appropriately large. Finally, we explain why the GD for an overparameterized deep linear network automatically avoids bad saddles.

## 1 Introduction

Deep linear neural networks, as a class of toy models, are frequently used to understand loss surfaces and gradient-based optimization methods related to non-convex problems. Dauphin et al. (2014) and Choromanska et al. (2015a) explored the loss function of deep nonlinear networks based on random matrix theory (such as a spherical spin-glass model). This theory essentially converts the loss surface of deep nonlinear neural networks into that of deep linear neural networks under certain assumptions, some of which are unrealistic. Choromanska et al. (2015b) suggested an open problem to establish a connection between the loss function of neural networks and the Hamiltonian of spherical spin-glass models under milder assumptions. Later, Kawaguchi (2016) successfully discarded most of these assumptions by analyzing the loss surface of the deep linear neural networks.

The landscape for deep linear neural network (Kawaguchi, 2016; Kawaguchi & Lu, 2017; Laurent & Brecht, 2018) focuses on several properties of the critical points: (i) every local minimum is a global minimum; (ii) every critical point that is not a local minimum is a saddle point; and (iii) there exists a saddle such that all eigenvalues of its Hessian are zeros if the network is deeper than three layers. Thus, for deep linear neural networks, convergence to a global minimum is impeded by the existence of poor saddles.

Lee et al. (2016) showed that the gradient method almost surely never converges to a strict saddle point, although the time cost can depend exponentially on the dimension (Du et al., 2017). Gradient descent (GD) with perturbations (Ge et al., 2015; Jin et al., 2017) can find a local minimizer in polynomial time. Thus, the trajectory approach combined with random initialization or random algorithm circumvents the obstacle of existence of poor saddles. According to studies on continuous time dynamics of a gradient flow (Du et al., 2018; Arora et al., 2018b), the balance property of deep linear network is preserved if the initialization is balanced. Arora et al. (2018a;b), Du & Hu (2019), and Hu et al. (2020) successfully proved that GD with its corresponding initialization schemes con-

verges to a global minimizer of deep linear neural networks with high probability. Furthermore, the rate of convergence is linear, and behaves like GD for a convex problem.

Hu et al. (2020) established that the convergence for Gaussian initialization can be very slow for deep linear neural networks with large depths, unless the width is almost linear. They also showed that orthogonal initialization in deep linear neural networks accelerate the convergence. Thus, the convergence behavior of the GD method, for training deep linear neural networks, crucially networks depends on the initialization.

Recent studies have demonstrated the connection between deep learning and kernel methods (Daniely, 2017; Arora et al., 2019a;b; Chizat et al., 2019; Lee et al., 2019; Du et al., 2019; Cao & Gu, 2019; Woodworth et al., 2020), especially the neural tangent kernel (NTK), introduced by Jacot et al. (2018). For most common neural networks, the NTK becomes constant (Jacot et al., 2018; Liu et al., 2020) and remains so throughout the training in the limit of a large layer width. Throughout the training, the neural networks are well described by their first-order Taylor expansion around their parameters at the initialization (Lee et al., 2019).

In this paper, we first evaluate the convergence region, i.e. the set of initialization parameters that lead to the linear convergence of GD for deep linear neural networks (see Lemma 4.1 or Lemma D.1). Next, we demonstrate that if the minimum width among all the hidden layers is sufficiently large, then the random initialization will fall into the convergence region with high probability (see Theorem 3.1, Theorem B.1, Theorem B.2 and Theorem B.3). Furthermore, the worst-case convergence rate of GD for deep linear neural networks is almost the same as the original convex problem with a corresponding learning rate. We also demonstrate that the GD trajectories for deep linear neural networks are arbitrarily close to those for the convex problem. The precise statement is related to remark 3, Theorem 3.2, Corollary 1 and Lemma 4.4 (also see Lemma D.5).

The present study was inspired by a recent reported work Du & Hu (2019); Hu et al. (2020), in which the authors carefully constructed the upper and lower bounds of the eigenvalues of the Gram matrix along the GD and established a linear convergence. In this paper, we generalize their results to strongly convex loss functions with layer varying widths and obtain sharper results. We also show that our rate of convergence for GD in deep linear neural networks is sharp in the sense that it matches the worst-case convergence rate for the original convex problem. The trajectories between the GD for deep linear neural networks and the original convex problem (1) can be arbitrary close. Furthermore, we show that if the width of each hidden layer is appropriately large, then the optimal rate does not depend on the random initialization types and network depth. Lastly, we elucidate the mechanism underlying the observed automatic avoidance of bad saddles by the GD for overparameterized deep linear networks.

## 2 PRELIMINARIES

### 2.1 PROBLEM SETUP

Let $x \in \mathbb{R}^{n_x}$ and $y \in \mathbb{R}^{n_y}$ be an input vector and a target vector, respectively. Define $\{(x_i, y_i)\}_{i=1}^m$ as a training dataset of size $m$, and let $X = [x_1, x_2, \cdots, x_m] \neq 0$ and $Y = [y_1, y_2, \cdots, y_m]$. Denote the weight parameters by $W \in \mathbb{R}^{n_y \times n_x}$.

Consider the well-studied convex optimization problem:

$$\underset{W}{\text{minimize}} \quad L(W) := \frac{1}{m} \sum_{i=1}^m l(Wx_i, y_i). \tag{1}$$

The GD for convex problem (1) with a learning rate of $\eta_*$ is given by:
$$W(t+1) = W(t) - \eta_* \nabla L(W(t)), t = 0, 1, 2, \cdots. \tag{2}$$

For any matrix $A$, let $\sigma_{max}(A)$ and $\sigma_{min}(A)$ be the largest and smallest singular values of $A$ respectively. Here, we consider two types of matrix norms and one type of semi-norm for $A$, $\|A\| := \sigma_{max}(A), \|A\|_F^2 := tr(AA^T)$, and $\|A\|_X := \|AP_X\|_F$, where $P_X = X(X^TX)^\dagger X^T$ is the orthogonal projection matrix onto the column space of $X$, and $(X^TX)^\dagger$ is the Moore–Penrose inverse.

For two real matrices $A, B$ with the same sizes, we consider their Frobenius inner product as well as their semi-inner product, $\langle A, B \rangle = \langle A, B \rangle_F := tr(A^T B), \langle A, B \rangle_X := \langle AP_X, BP_X \rangle$. Here, we list some basic properties for the semi-norm and semi-inner product.

**Lemma 2.1.** *The loss function $L(W)$ defined in (1) satisfies the following properties: for any $W, V \in \mathbb{R}^{n_y \times n_x}$,*

$1. L(W) = L(WP_X), \; 2. \nabla L(W) = \nabla L(WP_X)P_X, \; 3. \langle \nabla L(W), V \rangle_F = \langle \nabla L(W), V \rangle_X,$

$4. \|\nabla L(W)\|_F = \|\nabla L(W)\|_X, \; 5. \|W\|_X \equiv \|W\|_F$ *if and only if $X$ is full row rank.*

The next lemma demonstrates the importance of the semi-norm $\|\cdot\|_X$ in our analysis.

**Lemma 2.2.** *Assume that $l(\cdot, y)$ is $\alpha(l)-$strongly convex. Then, the following statements hold.*

1. *If $X$ is not a full row rank matrix, then $L(W)$ is neither strictly convex nor strongly convex with respect to $\|\cdot\|_F$.*

2. *$L(W)$ is $\frac{\alpha(l)\lambda_{min}(XX^T)}{m}-$strongly convex with respect to $\|\cdot\|_X$, where $\lambda_{min}(XX^T)$ is the smallest non-zero eigenvalue of $XX^T$.*

The proofs of the two aforementioned lemmas are provided in appendix A. Hereafter, if inner product is not specified, then we will consider the semi-inner product.

Assume that $L$ is $\alpha-$strongly convex ($\alpha > 0$), and $\nabla L$ is $\beta-$Lipschitz (with respect to the semi-norm $\|\cdot\|_X$); that means, for any $W, V \in \mathbb{R}^{n_y \times n_x}$,

$$L(W) \geq L(V) + \langle \nabla L(V), W - V \rangle_X + \frac{\alpha}{2} \|W - V\|_X^2,$$

$$\|\nabla L(W) - \nabla L(V)\|_X = \|\nabla L(W) - \nabla L(V)\|_F \leq \beta \|W - V\|_X.$$

Without loss of generality, we assume that $\alpha$ and $\beta$ are the best constants. Then, Lemma 2.2 implies that $\alpha \geq \frac{\alpha(l)\lambda_{min}(XX^T)}{m}$. Similarly, we can also show that $\beta \leq \frac{\beta(l)\lambda_{max}(XX^T)}{m}$, where $\nabla l(\cdot, y)$ is $\beta(l)-$Lipschitz and $\lambda_{max}(XX^T)$ is the largest eigenvalue of $XX^T$.

Define the effective condition number of the convex function $L$ by $\kappa = \kappa(L) = \frac{\beta}{\alpha} < \infty$. $\kappa$ appears naturally in the rate of convergence of the GD. Let $W_*$ be a global minimizer of $L(W)$, that is $L(W_*) = \min_W L(W)$. Notice that $W_*$ might not be unique, but $W_*P_X$ is unique.

The well-known results for the rate of convergence of GD (2) state are:

$$\eta_* = \frac{1}{\beta} \implies \mathcal{E}(t) \leq \left(1 - \frac{1}{\kappa}\right)^t \mathcal{E}(0), t = 1, 2, \cdots, \text{ as well as,} \tag{3}$$

$$\eta_* = \frac{2}{\alpha + \beta} \implies \mathcal{E}(t) \leq \frac{\beta}{2} \left(1 - \frac{4\kappa}{(1+\kappa)^2}\right)^t \|W(0) - W_*\|_X^2, t = 1, 2, \cdots, \tag{4}$$

where $\mathcal{E}(t) = L(W(t)) - L(W_*)$.

## 2.2 DEEP LINEAR NETWORK SETUP

Let $N - 1$ be the number of hidden layers. Assume $rank(X) = r$. Denote the weight parameters by $W_k \in \mathbb{R}^{n_k \times n_{k-1}}, k = 1, 2 \cdots, N$, with $n_N = n_y, n_0 = n_x$, where the $n_k$ is the width of the $k$-th layer. Set $n_{min} = \min\{n_1, n_2, \cdots, n_{N-1}\}$, and $n_{max} = \max\{n_1, n_2, \cdots, n_{N-1}\}$. For notational convenience, we denote $n_{j:i} = \prod_{i \leq k \leq j} n_k$ and denote $W_{j:i} = W_j W_{j-1} \cdots W_i$ for each $1 \leq i \leq j \leq N$. Define $n_{i-1:i} = 1$ and $W_{i-1:i} = I$ (of appropriate dimension) for completeness.

Considering the implicit regularization $W = W_{N:1}$ for the convex problem (1). We obtain the following non-convex optimization problem of deep linear neural networks:

$$\underset{W_1, \cdots, W_N}{\text{minimize}} \quad L(W_{N:1}) = \frac{1}{m} \sum_{i=1}^{m} l(W_{N:1}x_i, y_i). \tag{5}$$

**Example 2.1.** Specifically, if we set the loss to be $l(Wx_i, y_i) = \|Wx_i - y_i\|_2^2$, then $L(W) = \frac{1}{m} \|WX - Y\|_F^2$ is $\frac{2\lambda_{min}(XX^T)}{m}-$strongly convex, and $\nabla L$ is $\frac{2\lambda_{max}(XX^T)}{m}-$Lipschitz.

**Example 2.2.** Deep linear neural networks with regularization $\lambda \|W_N \cdots W_1 P_X\|_F^2$ can be converted into a new optimization problem

$$\underset{W_1, \cdots, W_N}{\text{minimize}} \quad L(W_{N:1}) + \lambda \|W_{N:1}\|_X^2.$$

Let $L_\lambda(W) = L(W) + \lambda \|W\|_X^2$. Then, $L_\lambda(\cdot)$ is $\alpha + 2\lambda$-strongly convex, and $\nabla L_\lambda(\cdot)$ is $\beta + 2\lambda$-Lipschitz.

More generally, if we consider regularization with a form $R(W) = \lambda \cdot g(W P_X)$, and $g(\cdot)$ is $\alpha'$-strongly convex, and $\beta'$-Lipschitz, then for the optimization problem

$$\underset{W_1, \cdots, W_N}{\text{minimize}} \quad L(W_{N:1}) + R(W_N \cdots W_1) =: L_R(W_N \cdots W_1),$$

we know that $L_R(\cdot)$ is $\alpha + \lambda\alpha'$-strongly convex, and $\nabla L_R(\cdot)$ is $\beta + \lambda\beta'$-Lipschitz.

## 2.3 INITIALIZATION SCHEMES

In previous studies, the following form of deep linear networks was considered, instead of (5):

$$\underset{W_1, \cdots, W_N}{\text{minimize}} \quad L(a_N W_{N:1}) = \frac{1}{m} \sum_{i=1}^m l(a_N W_{N:1} x_i, y_i), \tag{6}$$

where $a_N = 1/\sqrt{n_1 n_2 \cdots n_N}$ is a normalization constant.

By applying GD on (6), where we update $W_j$ simultaneously for $j$, we obtain

$$W_j(t+1) = W_j(t) - \eta \cdot a_N \left(W_{N:j+1}(t)\right)^T \nabla L\left(a_N W_{N:1}(t)\right) \left(W_{j-1:1}(t)\right)^T, j = 1, \cdots, N. \tag{7}$$

In a recent study, the authors considered GD (7) and adopted a Gaussian initialization (Du & Hu, 2019) or scaled orthogonal initialization (Hu et al., 2020) for initializing $W_j(0)$.

In this paper, we consider the following three kinds of random initializations, which generalize their idea.

**Gaussian initialization:** Let $W_1(0), \cdots, W_N(0)$ be the weight matrices at initialization. We assume that all the entries of $W_j, 1 \le j \le N$ are independent Gaussian random variables with a zero mean and unit variance.

Then, $a_N$ is a normalization constant in the sense that for any $x \in \mathbb{R}^{n_0}$, we have

$$\mathbb{E}\left[\|a_N W_{N:1}(0) x\|_2^2\right] = \|x\|_2^2. \tag{8}$$

In fact, all the initializations discussed in this paper satisfy (8).

*Remark* 1. Let $V_i(t) = \frac{1}{\sqrt{n_i}} W_i(t)$, for $1 \le i \le N$. Then, GD (7) with a unit variance Gaussian initialization is equivalent to

$$V_j(t+1) = V_j(t) - \frac{\eta}{n_j} \left(V_{N:j+1}(t)\right)^T \nabla L\left(V_{N:1}(t)\right) \left(V_{j-1:1}(t)\right)^T, \tag{9}$$

with a zero mean and variance $\frac{1}{n_i}$ Gaussian initialization for $V_i, i = 1, \cdots, N$.

GD (9) for loss (5) is equivalent to GD (7) for loss (6). Hereafter, we will only consider GD (7) for deep linear neural network (6).

**Orthogonal initialization:** We consider the so-called one peak random orthogonal projection and embedding initialization, which generalize the idea of orthogonal initialization (Hu et al., 2020).

**Definition 2.1.** An initialization $W_{N:1}(0) = W_N(0) W_{N-1}(0) \cdots W_1(0)$ is said to be a one peak random orthogonal projection and embedding initialization if there exists $1 \le p < N$, such that $n_0 \le n_1 \le n_2 \le \cdots \le n_p$, $n_p \ge n_{p+1} \ge n_{p+2} \ge \cdots n_{N-1} \ge n_N$, and $W_1(0), W_2(0), \cdots, W_p(0), W_{p+1}(0), W_{p+2}(0), \cdots, W_{n_N}(0)$ are independent and uniformly distributed over rectangular matrices, which satisfy

$$\begin{cases} W_i^T(0) W_i(0) = n_i I_{n_{i-1}}, 1 \le i \le p, \\ W_j(0) W_j^T(0) = n_{j-1} I_{n_j}, p+1 \le j \le N. \end{cases}$$

*Remark* 2. In this definition, $\sqrt{\frac{1}{n_i}} W_i(0), 1 \leq i \leq p$ are random embeddings and $\sqrt{\frac{1}{n_{j-1}}} W_j(0), p + 1 \leq j \leq N$ are random orthogonal projections. Notably, $A$ is a random orthogonal projection if and only if $A^T$ is a random embedding.

Arora et al. (2018a) studied the rate of convergence of GD to a global optimum for training a deep linear neural network for a balanced initialization. Here, we will consider a special case of balanced initialization, which is described as follows:

**Special balanced initialization:** Assume $n_1 = \cdots = n_{N-1} = n$. Consider the initialization $W_N(0) = \sqrt{n} U_N [I_{n_y}, 0_{n_y \times (n-n_y)}] V_N^T$, $W_1(0) = \sqrt{n} U_1 [I_{n_x}, 0_{n_x \times (n-n_x)}]^T V_1^T$ and $W_i(0) = \sqrt{n} U_i I_n V_i^T, 2 \leq i \leq N - 1$, where $U_{N-1}, U_N, V_1, V_i = U_{i-1}, 2 \leq i \leq N - 1$ are orthogonal matrices (random or deterministic), and $V_N$ has a uniform distribution over the orthogonal matrices. Notice that only $V_N$ is required to be random.

A simple estimation of the loss at the initialization is given by the following lemma.

**Lemma 2.3.** *If the initialization satisfies (8) for all $x$, then with probability at least $1 - \frac{\delta}{2}$, we have*

$$L(a_N W_{N:1}(0)) - L(W_*) \leq \beta B_\delta, \text{ where } B_\delta = \left( \frac{2 \cdot rank(X)}{\delta} + \|W_*\|_X^2 \right).$$

Note that the bound $B_\delta$ can be improved by using a sharp concentration inequality.

## 3 MAIN THEOREMS

Assume the thinnest layer is either the input layer or the output layer; that is $n_{min} \geq \max\{n_0, n_N\}$, and the ratio between the width of any hidden layer is bounded from above, precisely we have $\frac{n_{max}}{n_{min}} \leq C_0 < \infty$. The quantities $C_2, C_5$ and $C_6$ are defined in appendix D and are dependent on hyperparameters $n_N, \kappa, \delta, rank(X), C_0$, and $N$.

For notational convenience, we denote

$$\mathcal{E}(t) = L(W(t)) - L(W_*), \text{ and } \mathcal{E}_{DLN}(t) = L(a_N W_{N:1}(t)) - L(W_*).$$

Our assumptions and notation are now in place. We next state our main theorems in this section.

### 3.1 LINEAR CONVERGENCE OF DEEP LINEAR NEURAL NETWORKS

In appendix B we present a sharp estimate of the linear convergence of GD for deep linear neural networks in Theorem B.1 for Gaussian initialization, Theorem B.2 for orthogonal initialization, and Theorem B.3 for a special balanced initialization. In particular, with a specific learning rate $\eta = \frac{n_N}{\beta N}$, Theorem B.1 and Theorem B.2 yield the following optimum rate of convergence:

**Theorem 3.1.** *Given any $\delta, \varepsilon \in (0, \frac{1}{2})$, there exists a constant $C := C(\varepsilon)$, such that if one of the following two overparameterization condition holds:*

**1.** $n_{min} \geq C \cdot C_2 \cdot N$ *with the Gaussian initialization,*

**2.** $n_{min} \geq C \cdot C_5$ *with the one peak random orthogonal projection and embedding initialization*

*and with probability at least $1 - \delta$, then we have*

$$\mathcal{E}_{DLN}(t) \leq \left( 1 - \frac{1-\varepsilon}{\kappa} \right)^t \mathcal{E}_{DLN}(0), t = 1, 2, \cdots.$$

*Remark* 3. Consider GD (2) with a learning rate of $\eta_* = \frac{1}{\beta}$ and initialization $W(0) = a_N W_{N:1}(0)$. The well-known result of rate of convergence (3) for GD (2) of convex problem (1) matches the rates obtained from Theorem B.1 and Theorem B.2.

*Remark* 4. Du & Hu (2019), and Hu et al. (2020) showed that the number of iterations required to reach a precision $\varepsilon$ is $O\left(\kappa \log \frac{1}{\epsilon}\right)$ for $l_2$ loss. We only improved the rate of convergence and generalized their results to any strongly convex loss.

## 3.2 RESULTS OF TRAJECTORIES

Theorem 3.1 and remark 3 establish that the rate of convergence to a global optimum for GD to train a deep linear neural network is almost the same as the trajectories for the GD to train the corresponding convex problem with high probability, if the width is sufficiently large. Moreover, the GD for the fully-connected deep linear neural network (7) and that for GD (2) have almost the same trajectories.

Let $\eta_1 = \frac{2n_N}{\beta N}$ be an upper bound of the learning rate $\eta$. We can show that the trajectories of GD (7) for deep linear neural network (6) with a learning rate of $\eta < \eta_1$ are close to those of GD (2) with a learning rate of $\eta_* = \frac{N}{n_N}\eta$ for the corresponding convex problem (1) with high probability, if the width of each hidden layer is sufficiently large. The precise statement is as follows:

**Theorem 3.2.** *Consider the GD for deep linear neural network (7) with a learning rate of $\eta < \eta_1$ for $a_N W_{N:1}(t)$, $t = 0, 1, \cdots$, and GD (2) with a learning rate of $\eta_* = \frac{N}{n_N}\eta$ for $W(t)$, $t = 0, 1, \cdots$. Given $\tau, \delta \in (0, 1)$, there exists a constant $C := C(\tau, \eta/\eta_1)$ such that if one of the following three overparameterization conditions holds:*

1. $n_{min} \geq C \cdot C_2 \cdot N$ *with the Gaussian initialization,*

2. $n_{min} \geq C \cdot C_5$ *with the one peak random orthogonal projection and embedding initialization,*

3. $n_{min} \geq C \cdot C_6$ *with the special balanced initialization,*

*then with probability at least $1 - \delta$, we obtain*

$$\|a_N W_{N:1}(t) - W(t)\|_X^2 \leq D(\tau, q, t) \|a_N W_{N:1}(0) - W_*\|_X^2 , \tag{10a}$$

$$|\mathcal{E}_{DLN}(t) - \mathcal{E}(t)| \leq \beta \left( q^{t/2}\sqrt{D(\tau, q, t)} + \frac{1}{2}D(\tau, q, t) \right) \|a_N W_{N:1}(0) - W_*\|_X^2 , \tag{10b}$$

$$\mathcal{E}_{DLN}(t) \leq 3\beta(q + \tau)^t \|a_N W_{N:1}(0) - W_*\|_X^2 , \tag{10c}$$

*where $D(\tau, q, t) = \min \left\{ \frac{\tau}{1-q}, 2(q+\tau)^t \right\}$, with $0 < q < 1$ defined in (15).*

*Remark* 5. To the best of knowledge, this is the first paper that reveals that the trajectory of the overparameterized deep linear neural networks is close to the original convex problem with an appropriately rescaled learning rate.

**Corollary 1.** *According to Theorem 3.2, if we set $\eta = \frac{2n_N}{(\alpha+\beta)N}$, the following inequality holds with high probability,*

$$\mathcal{E}_{DLN}(t) \leq 3\beta \left( 1 - \frac{4\kappa}{(1+\kappa)^2} + \tau \right)^t \|a_N W_{N:1}(0) - W_*\|_X^2 . \tag{11}$$

*Notably, the rate of convergence in (11) is better than that in Theorem 3.1, because if $\kappa > 1$, then we can choose a sufficiently small $\tau$ such that the following inequality holds:*

$$1 - \frac{4\kappa}{(1+\kappa)^2} + \tau < 1 - \frac{1}{\kappa}.$$

Theorem 3.1, Theorem 3.2, Theorem B.1, and Theorem B.2 indicate that the implicit regularization induced by the GD for a convex problem recovers the convex problem itself in terms of optimization, at the cost of linear convergence only with high probability for random initialization.

*Remark* 6. Recall the constants $C_2, C_5$, and $C_6$ defined in appendix B. The term $\frac{rank(X)}{\delta}$ is not optimal, since our concentration inequality depends only on the second moment. By using stronger concentration inequalities for our Lemma 2.3, similar to the proof of proposition 6.5 (Du & Hu, 2019) and Lemma 4.2 (Hu et al., 2020), the $\frac{rank(X)}{\delta}$ can be improved to $1 + \log(\frac{rank(X)}{\delta})$. $C_2$ is proportional to $\kappa^2$, which is slightly better than the constant in Du & Hu (2019), which is proportional to $\kappa^3$. $C_5$ is also slightly better than the constant reported by Hu et al. (2020), since we do not have the extra term $\frac{\|X\|_F^2}{\|X\|^2}$. The improvement of the constant is mainly due to the introduction of the semi-norm $\|\cdot\|_X$.

## 4  INSIGHTS FOR THEOREM 3.2

**Initialization and convergence region:** Arora et al. (2018a) showed that if the initialization is approximately balanced, and the product matrix $W_{N:1}(0)$ is very close to a global minimizer, then the GD linearly converges to the global minimum for the deep linear network without any width requirement. However, the convergence region in (Arora et al., 2018a) is very small, because $W_{N:1}(0)$ needs to be very close to $W_*$. Later, Du & Hu (2019), and Hu et al. (2020) successfully proved that the GD with a Gaussian, or orthogonal initialization linearly converges to a global minimizer of the overparameterized deep linear neural network with high probability. They introduced a technique to analyze the trajectories of GD with large widths for any deterministic initialization.

We introduce the following lemma, which describes the linear convergence result for a deep linear network with a deterministic initialization.

**Lemma 4.1.** *Under the setting of Lemma D.1, the GD for a deep linear network satisfies*

$$\mathcal{E}_{DLN}(t) \leq (1 - \eta\gamma)^t \mathcal{E}_{DLN}(0), t = 1, 2, \cdots.$$

Our convergence region (see (31) in Lemma D.1 and Definition D.1) originates from the analysis of Du & Hu (2019), and Hu et al. (2020) and can be view as a neighborhood of the special balanced initialization, if $n_1 = n_2 = \cdots = n_{N-1}$. Both Gaussian and orthogonal initialization are approximately balanced.

For $l_2$ loss, without loss of generality, we can assume $X$ to be a full rank matrix and $L(W_*) = 0$ because of the decomposition method in claim B.1 from Du & Hu (2019). However, when considering a general strongly convex loss, we have to confront the low rank $X$ directly in our analysis. Thus, $\|\cdot\|_X$ appears naturally and aids in achieving the sharp rate of convergence in our main theorems. In addition to the technique reported in Du & Hu (2019), and Hu et al. (2020), we also used classical convex optimization techniques (such as inequalities in Lemma C.1, and Polyak-Łojasiewicz inequality in (26)) as well as the classical concentration inequalities for beta distribution (such as the Chernoff type bound in Lemma F.3).

**Why GD trajectories for overparameterized deep linear neural networks with approximate balanced initialization are close to those for convex problems?** The underlying mechanism can be understood as follows: Even though recent results of (Ziyin et al., 2022) can describe the exact global minimizer for a deep linear network (with a regularization term such as $l_2$), the evolution of each $W_j$ is still difficult to track. Instead, we consider the discrete dynamics for product matrices (see (41) and (42)):

$$a_N W_{N:1}(t+1) = a_N W_{N:1}(t) - \eta \cdot P(t)[\nabla L(a_N W_{N:1}(t) P_X)] + a_N E(t).$$

For their own linear operator $P_t$, Du & Hu (2019) showed that $\lambda_{max}(P_t) \leq O(\frac{N}{n_N}) \cdot \lambda_{max}(X^T X)$ and $\lambda_{min}(P_t) \geq \Omega(\frac{N}{n_N}) \cdot \lambda_{min}(X^T X)$. To the best of our knowledge, the present paper is the first to proved that for our operator $P(t)[\cdot] \approx \frac{N}{n_N} I$ (also see (44)), where $I$ is the identity operator. $E(t)$ is negligible, which leads to the following result on discrete dynamics (see Lemma D.3).

**Lemma 4.2.** *Under the setting of Lemma D.3, we have*

$$a_N W_{N:1}(t+1) = a_N W_{N:1}(t) - \frac{N}{n_N} \eta \nabla L(a_N W_{N:1}(t)) + R(t),$$

*with* $\|R(t)\|_X \leq \tau \|a_N W_{N:1}(t) - W_*\|_X$.

Without the $R(t)$ term, the discrete dynamics is exactly the GD for a convex function. To control the distance between the two trajectories, we introduce the following lemma (also see Lemma D.4).

**Lemma 4.3.** *Assume $\tau \in [0, 1)$, and consider a discrete dynamical system $V(t)$ such that,*

$$V(t+1) = V(t) - \eta_* \nabla L(V(t)) + R(t), \text{ where } \|R(t)\|_X \leq \tau \|V(t) - W_*\|_X.$$

*If $\eta_* \leq 2/\beta$, then we have $\|V(t) - W_*\|_X^2 \leq (q + 7\tau)^t \|V(0) - W_*\|_X^2$, where $q$ is defined in (15).*

With the help of this lemma, we further obtain the following trajectories comparison lemma (also see Lemma D.5), which leads to the main conclusions in Theorem 3.2.

**Lemma 4.4.** *Under the setting of Lemma D.5, we have*

$$\|a_N W_{N:1}(t) - W(t)\|_X^2 \le D(\tau, q, t) \|a_N W_{N:1}(0) - W_*\|_X^2, \tag{12a}$$

$$|\mathcal{E}_{DLN}(t) - \mathcal{E}(t)| \le \beta \left( q^{t/2} \sqrt{D(\tau, q, t)} + \frac{1}{2} D(\tau, q, t) \right) \|a_N W_{N:1}(0) - W_*\|_X^2, \tag{12b}$$

$$\mathcal{E}_{DLN}(t) \le 3\beta(q + \tau)^t \|a_N W_{N:1}(0) - W_*\|_X^2, \tag{12c}$$

*where $D(\tau, q, t) = \min\left\{ \frac{\tau}{1-q}, 2(q+\tau)^t \right\}$, with $0 < q < 1$ defined in (15).*

**Why do bad saddles not affect GD for overparameterized deep linear neural networks?** A critical point $x^*$ of $f$ is a bad saddle if $\lambda_{min}(\nabla^2 f(x^*)) = 0$. Kawaguchi (2016) showed that deep linear networks have bad saddles, and thus, in general, a vanishing Hessian can hinder the optimization. Theorem 2.3 in Kawaguchi (2016) explains that for all bad saddles satisfy that $W_{N-1:2}$ is a non-full rank matrix. Thus, to show that the trajectories of GD are away from bad saddle points, it is sufficient to demonstrate that $\inf_t \sigma_{min}(W_{N-1:2}(t)) > 0$. According to previous studies, there are two main ways to avoid bad saddles for GD to train deep linear networks.

On the one hand, following Arora et al. (2018b), it can be showed that if the approximate balanced initialization satisfies $\|W_{N:1}(0) - W_*\|_F \le \sigma_{min}(W_*) - c$, for some $0 < c < \sigma_{min}(W_*)$, then $\sigma_{min}(W_{N:1}(t)) \ge c$ through the training as well as $\|W_1(t)\| \le (4\|W_*\|_F)^{1/N}$, and $\|W_N(t)\| \le (4\|W_*\|_F)^{1/N}$, then $\sigma_{min}(W_{N-1:2}(t)) \ge \frac{\sigma_{min}(W_{N:1}(t))}{\|W_1(t)\|\|W_N(t)\|} \ge \frac{c}{(4\|W_*\|)^{2/N}}$.

On the other hand, if we assume that our rescaled and overparameterized weight initialization falls into convergence region (31), then we can show that (see $\mathcal{B}(t)$ in the proof of Lemma D.1)

$$\sigma_{min}(W_{N-1:2}(t)) \ge \max\left\{ \frac{\sigma_{min}(W_{N:2}(t))}{\sigma_{max}(W_N(t))}, \frac{\sigma_{min}(W_{N-1:1}(t))}{\sigma_{max}(W_1(t))} \right\}.$$

Thus, $\sigma_{min}\left(\frac{W_{N-1:2}}{(n_{N-1:2})^{1/2}}\right) \ge e^{-c_1-c_2} \max\{\frac{n_1}{n_{N-1}}, \frac{n_{N-1}}{n_1}\} \ge e^{-c_1-c_2} > 0$.

In conclusion, we first made a conjecture that according to Arora et al. (2018b), for a non-overparameterized deep linear network, there are no bad saddles satisfying $\|W_{N:1}(0) - W_*\|_F < \sigma_{min}(W_*)$. Thus, $\|W_{N:1}(0) - W_*\|_F < \sigma_{min}(W_*)$ is indeed a convergence region. However, this region in general is very small, and can even be empty if $\sigma_{min}(W_*) = 0$. For an overparameterized deep linear network, the GD initialized in the convergence region will force the trajectories away from all the bad saddles.

**Why does the width have to be large?** We will discuss overparameterization phenomena in deep linear networks. For simplicity, we consider a special balanced initialization. First, we know that $\|W_i(t) - W_i(0)\|_F = O(\frac{1}{N})$ (see $\mathcal{C}(t)$ through the proof of Lemma D.1), provided $\eta = h\frac{n_N}{\beta N}$ and $\gamma = O(\frac{\alpha N}{n_N})$, where $h \in (0, 2)$.

An overparameterized deep linear network around the special balanced initialization is full of global minimizers, i.e., the trajectory limit $(W_1^*, W_2^*, \cdots, W_N^*)$ is in the $O(\frac{1}{N})$ neighborhood of the special balanced initialization $(W_1(0), W_2(0), \cdots, W_N(0))$. Notably,

$$\sigma_{min}(W_{N-1:2}(0)) = \prod_{j=2}^{N-1} \sigma_{min}(W_j(0)) = \sigma_{max}(W_{N-1:2}(0)) = \prod_{j=2}^{N-1} \sigma_{max}(W_j(0)) = n^{(N-2)/2},$$

as well as for any $(W_1, W_2, \cdots, W_N)$ in the $O(\frac{1}{N})$ neighborhood of any given initialization $(W_1(0), W_2(0), \cdots, W_N(0))$, we have (detailed argument can be found in the proof of $\mathcal{B}(t)$ in Lemma D.1):

$$\|W_{N-1:2} - W_{N-1:2}(0)\|_F \le \sum_{s=1}^{N-2} \binom{N-1}{s} O(\frac{1}{N})^s (n^{(N-2-s)/2}) \le n^{(N-2)/2} O\left(\frac{1}{\sqrt{n}}\right).$$

Thus in terms of landscape, we have

$$\frac{\sigma_{min}(W_{N-1:2})}{\sigma_{min}(W_{N-1:2}(0))} \ge \frac{\sigma_{min}(W_{N-1:2}(0)) - \|W_{N-1:2} - W_{N-1:2}(0)\|_F}{\sigma_{min}(W_{N-1:2}(0))} \ge 1 - O(\frac{1}{\sqrt{n}}), \tag{13}$$

which implies that no bad saddle is present in the $O(\frac{1}{N})$ neighborhood of the special balanced initialization, if the width $n$ is sufficiently large.

In terms of training, we have (see the proof of Lemma D.1),

$$\|W_i(t) - W_i(0)\|_F = O(\frac{1}{N}), \ \|W_i(0)\|_F = n, 2 \leq i \leq N-1,$$

as well as $\frac{\|W_i(t)-W_i(0)\|_F}{\|W_i(0)\|_F} = O(\frac{1}{Nn})$. Thus for an overparameterized deep linear network, the GD with an approximate balanced initialization only trains $W_1$ and $W_N$, and the other weight matrices remain almost constant. Here, we provide empirical evidence in appendix G to support the aforementioned argument.

On the other hand, the sharp rate of convergence depends on the trajectory limit, and when the minimum width is sufficiently large, the trajectory limit and the initialization are not far away from each other. For deep linear network with small widths, the result (Ziyin et al., 2022) might shed light on convergence analysis, because the exact global minimizer can be described for a deep linear network with $L_2$ regularization.

**Numerical Experiments:** In appendix H, we will discuss some empirical evidence to support the main results shown in Section 3. Further, Figure 1 and 2 in appendix H show plots of the logarithm of loss as a function of number of iterations. When $n$ is small, the trajectories of loss for deep linear neural networks do not decrease in some iterations. However, when $n$ is large, the loss trajectories are close to those for the corresponding convex problem.

## 5 OVERVIEW OF THE PROOFS OF MAIN THEOREMS AND LEMMAS

In this section, we provide an overview of the proofs for all the theorems obtained in the main results. Since Theorem 3.1 in the main results is a special cases of general theorems with non-optimal learning rates (see Theorem B.1 and Theorem B.2), we only need to focus on the proofs of the general theorems (see Theorem B.1, Theorem B.2, Theorem B.3, and Theorem 3.2).

We begin with the convergence region of deep linear neural networks, which is basically the set of initializations that lead to the convergence of the GD for deep linear neural networks. The precise definition can be found in appendix D. Lemma 4.1 and Lemma 4.4 (also see Lemma D.1 and Lemma D.5) prove that this convergence region satisfies the following properties: if the initialization falls into the convergence region, then

(i) the GD is guaranteed to converge to a global minimizer of the deep linear neural networks,

(ii) the worst-case GD rate of convergence for the deep linear neural networks, which is a non-convex problem, is almost the same as the corresponding convex problem with a corresponding learning rate, and,

(iii) the trajectories of the GD for the deep linear neural networks are arbitrarily close to those for the corresponding convex problem.

More precisely, Lemma 4.1 (also see Lemma D.1) establishes the convergence region for a deterministic initialization, and it demonstrates the first two properties, (i) and (ii). Additionally, in appendix E and appendix F we also prove that the spectral properties of the products of random matrices partially reveal that the overparameterization realized by adding the width of each hidden layer guarantees that the random initialization falls into the convergence region with high probability. These results provide a foundation to establish the main linear convergence theorem for random initialization (see Theorem B.1, B.2, and B.3).

By contrast, Lemma 4.2 (also see Lemma D.3) shows that if the initialization falls into the convergence region, then the update rule for the product of weight matrices in the GD for deep linear neural networks is more or less given by (2). This result can be used to establish both Lemma 4.4 (also see Lemma D.5), and Theorem 3.2, which is precisely property (iii) of the convergence region for deterministic and non-deterministic initializations, respectively.

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

## A  PROOFS OF BASIC PROPERTIES OF THE SEMI-NORM

*Proof of Lemma 2.1.* The first property is a direct consequence of the definition of the projection matrix $P_X$.

Notice that

$$\frac{1}{\varepsilon}(L(W + \varepsilon\Delta W) - L(W)) = \frac{1}{\varepsilon}(L(WP_X + \varepsilon\Delta WP_X) - L(WP_X)).$$

Considering $\varepsilon \to 0$, the definition of the directional derivative implies

$$\langle \nabla L(W), \Delta W \rangle_F = \langle \nabla L(WP_X), \Delta WP_X \rangle_F = \langle \nabla L(WP_X)P_X, \Delta W \rangle_F, \forall \Delta W \in \mathbb{R}^{n_y \times n_x},$$

since $P_X = P_X^T$. This completes the proof of the second property.

The third property is derived from the condition: orthogonal projection matrix satisfies $P_X = P_X^T = P_X^2 = P_X^3$, since

$$\langle \nabla L(W), V \rangle_F = \langle \nabla L(WP_X)P_X, V \rangle_F$$
$$= \langle \nabla L(WP_X)P_X^2, VP_X \rangle_F = \langle \nabla L(WP_X)P_X, V \rangle_X = \langle \nabla L(W), V \rangle_X.$$

If we set $V = \nabla L(W)$, then the fourth property is implied by the third property.

For the last property, first recall that $\|W\|_X = \|WP_X\|_F$ and $P_X = X(X^TX)^\dagger X^T$. $X$ is of a full row rank matrix if and only if $P_X$ is an identity matrix, which completes the proof.

$\square$

*Proof of Lemma 2.2.* Because $X$ is not full row rank, we know that $I - P_X \neq 0$. There exists $W$ such that $W(I - P_X) \neq 0$. Applying the first property in Lemma 2.1, we obtain

$$L(\frac{1}{2}W + \frac{1}{2}WP_X) = L((\frac{1}{2}W + \frac{1}{2}WP_X)P_X) = L(WP_X) = \frac{1}{2}L(W) + \frac{1}{2}L(WP_X),$$

provided $W \neq WP_X$.

Hence, $L$ is not strictly convex, which implies that $L$ is not strongly convex.

To prove the second property, it is sufficient to show that $g(W) = L(W) - \frac{\alpha(l)\lambda_{min}(XX^T)}{m}\|W\|_X^2$ is convex. Obviously,

$$g(W) = L(W) - \frac{\alpha(l)}{m}\sum_{i=1}^m \|Wx_i - y_i\|_2^2 + \frac{\alpha(l)}{m}(\|WX - Y\|_F^2 - \lambda_{min}(X^TX)\|W\|_X^2). \quad (14)$$

$L(W) - \frac{\alpha(l)}{m}\sum_{i=1}^m \|Wx_i, y_i\|_F^2$ is convex, because $l(\cdot, y_i)$ is strongly convex. The Hessian of $\|WX - Y\|_F^2 - \lambda_{min}(W^TW)\|WP_X\|_F^2$ has no negative eigenvalue; thus the second term in (14) is also convex. This completes the proof. $\square$

## B  EXACT STATEMENTS OF THE MAIN THEOREMS

Definitions of some quantities:

$$q = \begin{cases} 1 - \alpha\eta_*(2 - \eta_*\alpha), & 0 < \eta_* \leq \frac{2}{\alpha+\beta} \\ 1 - \beta\eta_*(2 - \eta_*\beta), & \frac{2}{(\alpha+\beta)} < \eta_* < \frac{2}{\beta}, \end{cases} \quad (15)$$

$$B_\delta = \left( \frac{2 \cdot rank(X)}{\delta} + \|W_*\|_X^2 \right),$$

$$C_1 = n_N \kappa^2 B_\delta \frac{C_0}{(\eta_0 - \eta)^2/\eta_0^2} + \ln N,$$

$$C_2 = n_N \kappa^2 B_\delta C_0 + \ln N,$$

$$C_3 = n_N \kappa^2 B_\delta \frac{C_0}{(\eta_0 - \eta)^2/\eta_0^2} + C_0 \ln(\underline{N}),$$

$$C_4 = n_N \kappa^2 B_\delta \frac{1}{(\eta_0 - \eta)^2/\eta_0^2},$$

$$C_5 = n_N \kappa^2 B_\delta C_0 + C_0 \ln(\underline{N}),$$

$$C_6 = n_N \kappa^2 B_\delta,$$

where $\underline{N}$ denotes the number of distinct elements in the set $\{n_1, \cdots, n_{N-1}\}$, $\eta_1 = \frac{2n_N}{N\beta}$, and $\eta_0 = \frac{2n_N}{e^{2c}N\beta}$ with $c > 0$.

**Theorem B.1.** *Given any $c > 0$, and $0 < \delta < 1/2$, define $\eta_0 = \frac{2n_N}{e^{2c}N\beta}$, and consider the learning rate of $\eta < \eta_0$. There exists a constant $C := C(c)$, such that if*

$$n_{min} \geq C \cdot C_1 \cdot N, \tag{16}$$

*then with probability at least $1 - \delta$ over the random Gaussian initialization, we have*

$$\mathcal{E}_{DLN}(t) \leq \left( 1 - 4e^{-c\frac{\frac{\eta}{\eta_0}(1 - \frac{\eta}{\eta_0})}{\kappa}} \right)^t \mathcal{E}_{DLN}(0).$$

**Theorem B.2.** *Given any $c > 0$, and $0 < \delta < 1/2$, define $\eta_0 = \frac{2n_N}{e^{2c}\beta N}$, and consider the learning rate to be $\eta < \eta_0$. There exists a constant $C := C(c)$, such that if*

$$n_{min} \geq C \cdot C_3, \tag{17}$$

*then with probability at least $1 - \delta$ over the random one peak projection and embedding initialization, we have*

$$\mathcal{E}_{DLN}(t) \leq \left( 1 - 4e^{-c\frac{\frac{\eta}{\eta_0}(1 - \frac{\eta}{\eta_0})}{\kappa}} \right)^t \mathcal{E}_{DLN}(0).$$

*Specially, if $n_1 = n_2 = \cdots = n_{N-1} = n \geq \min\{n_N, n_0\}$, then requirement (17) can be replaced by*

$$n \geq C \cdot C_4. \tag{18}$$

*Remark* 7. Assume $L(a_N W_N \cdots W_1) = \frac{1}{2} \|a_N W_N \cdots W_1 X - Y\|_F^2$, and $n_1 = \cdots = n_{N-1} = n$. Then, for Gaussian initialization, our Theorem B.1 leads to Theorem 4.1 in Du & Hu (2019). Similarly, for orthogonal initialization, our Theorem B.2 leads to Theorem 4.1 of Hu et al. (2020).

Next, we present a version of the theorem related to balanced initialization.

**Theorem B.3.** *Assume $n_1 = \cdots = n_{N-1} = n$. Given any $c > 0$, and $0 < \delta < 1/2$, define $\eta_0 = \frac{2n_N}{e^{2c}\beta N}$, and consider the learning rate as $\eta < \eta_0$. There exists a constant $C := C(c)$, such that as long as*

$$n \geq C \cdot C_4. \tag{19}$$

*then with probability at least $1 - \delta$ over the special balanced initialization, we have*

$$\mathcal{E}_{DLN}(t) \leq \left( 1 - 4e^{-c\frac{\frac{\eta}{\eta_0}(1 - \frac{\eta}{\eta_0})}{\kappa}} \right)^t \mathcal{E}_{DLN}(0).$$

## C INEQUALITIES IN CONVEX OPTIMIZATION

Convex optimization has been studied for about a century. Recall the definitions and basic inequalities for $\alpha-$strongly convex and $\beta-$Lipschitz functions.

**Definition C.1.** A continues differentiable function $f$ is said to be $\beta-$ Lipschitz if the gradient $\nabla f$ is $\beta-$ Lipschitz, that is if for all $x, y$,

$$\|\nabla f(y) - \nabla f(x)\| \leq \beta \|y - x\|, \tag{20}$$

$f$ is said to be $\alpha-$strongly convex if for all $x, y$, we have

$$f(y) \geq f(x) + \langle \nabla f(x), y - x \rangle + \frac{\alpha}{2} \|y - x\|^2. \tag{21}$$

**Proposition C.1.** If $f$ is $\alpha-$strongly convex and $\nabla f$ is $\beta-$Lipschitz with respect to a (semi-)norm, then $\alpha \leq \beta$ and

$$\langle \nabla f(x), y - x \rangle + \frac{\alpha}{2} \|y - x\|^2 \leq f(y) - f(x) \leq \langle \nabla f(x), y - x \rangle + \frac{\beta}{2} \|y - x\|^2, \tag{22}$$

$$\langle \nabla f(x) - \nabla f(y), x - y \rangle \geq \frac{\alpha\beta}{\alpha + \beta} \|x - y\|^2 + \frac{1}{\alpha + \beta} \|\nabla f(x) - \nabla f(y)\|^2, \tag{23}$$

$$\|\nabla f(x) - \nabla f(y)\| \geq \alpha \|x - y\|, \tag{24}$$

$$f(x) - f(y) \leq \langle \nabla f(x), x - y \rangle - \frac{1}{2\beta} \|\nabla f(x) - \nabla f(y)\|^2. \tag{25}$$

*Proof of Proposition C.1.* We only prove the last inequality.
Let $z = y - \frac{1}{\beta}(\nabla f(y) - \nabla f(x))$. Since $f$ is convex $\beta-$Lipschitz, we have

$$f(z) - f(x) \geq \langle \nabla f(x), z - x \rangle$$

and

$$f(z) - f(y) \leq \langle \nabla f(y), z - y \rangle + \frac{\beta}{2} \|z - y\|^2.$$

Thus,

$$\begin{aligned} f(x) - f(y) &= f(x) - f(z) + f(z) - f(y) \\ &\leq \langle \nabla f(x), x - z \rangle + \langle \nabla f(y), z - y \rangle + \frac{\beta}{2} \|z - y\|^2 \\ &= \langle \nabla f(x), x - y \rangle - \frac{1}{2\beta} \|\nabla f(x) - \nabla f(y)\|^2. \end{aligned}$$

$\square$

Before we prove Lemma D.1, let us first include and prove the following result.

**Lemma C.2.** *1. Assume $L$ is $\alpha-$strongly convex, $\alpha > 0$. Denote a global minimizer of $L$ by $W_*$. Then, for any $W$,*

$$L(W_*) - L(W) \geq -\frac{1}{2\alpha} \|\nabla L(W)\|_X^2. \tag{26}$$

*2. Assume $\nabla L$ is $\beta-$Lipschitz, then*

$$L(W_*) - L(W) \leq -\frac{1}{2\beta} \|\nabla L(W)\|_X^2. \tag{27}$$

*Proof of Lemma C.2.* **1.** First, we know that $\nabla L(W_*) = 0$. $L$ is $\alpha-$strongly convex, which implies the inequality (22) holds. Thus

$$L(V) - L(W) \geq \langle \nabla L(W), V - W \rangle_X + \frac{\alpha}{2} \|V - W\|_X^2 =: g(V).$$

Minimizing both sides in terms of $V$ gives (26).

Now we focus on minimizing $g(V)$. Since $g(V) \in C^1$ and the global minimizer exits, we have
$$\nabla g(V^*) = \nabla L(W)P_X + \alpha(V^* - W)P_X = 0,$$
where $V^*$ is a global minimizer for $g(V)$. Thus,
$$g(V^*) = -\frac{1}{2\alpha} \|\nabla L(W)\|_X^2. \tag{28}$$

**2.** Applying proposition C.1 to a $\beta-$Lipschitz function $\nabla L$, we obtain
$$L(W_*) - L(W)$$
$$\leq \langle \nabla L(W_*), W_* - W \rangle_X - \frac{1}{2\beta} \|\nabla L(W) - \nabla L(W_*)\|_X^2$$
$$= -\frac{1}{2\beta} \|\nabla L(W)\|_X^2.$$

$\square$

# D  CONVERGENCE REGION

In this section, we evaluate a class of the convergence region for deep linear neural networks with a deterministic initialization. Define $A|_{\mathcal{R}(X)} = AX^T(XX^T)^- X = AP_X$, and view $A|_{\mathcal{R}(X)}$ as a linear operator on $\mathcal{R}(X)$.

Recall the optimization problem
$$\underset{W_1,\cdots,W_N}{\text{minimize}} \quad L^N(W_1, \cdots W_N) := \frac{1}{m} \sum_{i=1}^m l(a_N W_{N:1} x_i, y_i) = L(a_N W_{N:1}), \tag{29}$$

and GD
$$\begin{cases} W_j(t+1) = W_j(t) - \eta \frac{\partial L^N}{\partial W_j}(W_1(t), \cdots, W_N(t)), j = 1, \cdots, N, \\ \text{where } \frac{\partial L^N}{\partial W_j}(W_1, \cdots, W_N) = a_N(W_{N:j+1})^T \nabla L(a_N W_{N:1})(W_{j-1:1})^T, \end{cases} \tag{30}$$
where the normalization factor $a_N = \frac{1}{\sqrt{n_1 n_2 \cdots n_{N-1} n_N}}$.

The following theorem generalizes the idea from a recent work (Du & Hu, 2019; Hu et al., 2020).

For notational convenience, we denote $W_{j:i}(t) = W_j(t) \cdots W_i(t)$, $L_t = L(a_N W_{N:1}(t))$, $\nabla L_t = \nabla L(a_N W_{N:1}(t))$ etc.

**Lemma D.1.** *Assume the initialization simultaneously satisfies the following conditions:*
$$\begin{cases} \sigma_{max}(W_{N:i+1}(0)) \leq e^{c_1/2}(n_{N-1:i})^{1/2}, 1 \leq i \leq N-1, \\ \sigma_{min}(W_{N:i+1}(0)) \geq e^{-c_2/2}(n_{N-1:i})^{1/2}, 1 \leq i \leq N-1, \\ \sigma_{max}(W_{i-1:1}(0)|_{\mathcal{R}(X)}) \leq e^{c_1/2}(n_{i-1:1})^{1/2}, 2 \leq i \leq N, \\ \sigma_{min}(W_{i-1:1}(0)|_{\mathcal{R}(X)}) \geq e^{-c_2/2}(n_{i-1:1})^{1/2}, 2 \leq i \leq N, \\ \|W_{j:i}(0)\| \leq M/2 \cdot N^\theta (\prod_{i \leq k \leq j-1} n_k \cdot \max\{n_{i-1}, n_j\})^{1/2}, 1 < i \leq j < N, \\ L_0 - L(W_*) \leq \beta B_0 =: B, \end{cases} \tag{31}$$
*where $c_1, c_2, M$ are positive constant and $\theta \geq 0$.*
*Notice that $B_0$ is a proper upper bound for $\|a_N W_{N:1}(0)\|_X^2 + \|W_*\|_X^2$.*

*Set the learning rate as $\eta = \frac{(1-\varepsilon)2n_N}{e^{6c_1+3c_2}\beta N}$, where $0 < \varepsilon < 1$. Define $\gamma = \frac{2e^{6c_1}\varepsilon\alpha N}{n_N}$.*

*Assume that*
$$n_{min} \geq \frac{C(c_1, c_2)M^2\kappa^2 B_0}{\varepsilon^2} N^{2\theta} n_N. \tag{32}$$

*Then, GD (30) satisfies*
$$L_t - L(W_*) \leq (1 - \eta\gamma)^t (L_0 - L(W_*)), t = 1, 2, \cdots.$$

**Definition D.1.** For given $c_1, c_2, M, B_0 > 0$, and $\theta \geq 0$, we define the convergence region $\mathcal{R}(c_1, c_2, \theta, M, B_0)$ by the set of initialization that satisfies the inequality system (31).

*Remark* 8. The condition (31) describes the convergence region for initialization and the condition (32) describes the overparameterization for deep linear neural networks. At this time, it is not clear how large this convergence region is. Later, we will show that the properly scaled random initialization with some extra mild overparameterization conditions will fall into this convergence region with high probability.

*Proof of Lemme D.1.* To prove Lemma D.1, it suffices to show that the following three properties hold $\mathcal{A}(t), \mathcal{B}(t)$, and $\mathcal{C}(t)$ for all $t = 0, 1, \cdots$.

1. $\mathcal{A}(t)$:
$$L_t - L(W_*) \leq (1 - \eta\gamma)^t (L_0 - L(W_*)).$$

2. $\mathcal{B}(t)$:
$$\begin{cases} \sigma_{max}(W_{N:i+1}(t)) \leq e^{c_1}(n_{N-1:i})^{1/2}, 1 \leq i \leq N-1, \\ \sigma_{min}(W_{N:i+1}(t)) \geq e^{-c_2}(n_{N-1:i})^{1/2}, 1 \leq i \leq N-1, \\ \sigma_{max}(W_{i-1:1}(t)|_{\mathcal{R}(X)}) \leq e^{c_1}(n_{i-1:1})^{1/2}, 2 \leq i \leq N, \\ \sigma_{min}(W_{i-1:1}(t)|_{\mathcal{R}(X)}) \geq e^{-c_2}(n_{i-1:1})^{1/2}, 2 \leq i \leq N, \\ \|W_{j:i}(t)\| \leq M \cdot N^\theta(\frac{1}{n_{min}} \prod_{i-1 \leq k \leq j} n_k)^{1/2}, 1 < i \leq j < N. \end{cases}$$

3. $\mathcal{C}(t)$:
$$\|W_i(t) - W_i(0)\|_F \leq \frac{2e^{2c_1}\sqrt{2\beta B}}{\sqrt{n_N}\gamma} =: R, 1 \leq i \leq N.$$

Using simultaneous induction, the proof of Lemma D.1 is divided into the following three claims.

*Claim* 1. $\mathcal{A}(0), \cdots, \mathcal{A}(t), \mathcal{B}(0), \cdots, \mathcal{B}(t) \implies \mathcal{C}(t+1)$.

*Claim* 2. $\mathcal{C}(t) \implies \mathcal{B}(t)$, if $n_{min} \geq \frac{C(c_1, c_2)M^2\kappa^2 B_0}{\varepsilon^2}N^{2\theta}n_N$, where $C(c_1, c_2)$ is a positive constant only depend on $c_1, c_2$.

*Claim* 3. $\mathcal{A}(t), \mathcal{B}(t) \implies \mathcal{A}(t+1)$, if $n_{min} \geq C(c_1, c_2)M^2 B_0 N^{2\theta}n_N$, where $C(c_1, c_2)$ is a positive constant only depend on $c_1, c_2$.

$\square$

*Proof of Claim 1.* As a consequence of Lemma C.2 and Lemma 2.1, and $\mathcal{A}(s)$, $s \leq t$, we have
$$\begin{aligned} \|\nabla L(a_N W_{N:1}(s))\|_F^2 &= \|\nabla L_s - \nabla L(W_* P_X)\|_X^2 \\ &\leq 2\beta[L_s - L(W_*)] \\ &\leq 2\beta(1 - \eta\gamma)^s B. \end{aligned} \tag{33}$$

From $\mathcal{A}(0), \cdots, \mathcal{A}(t), \mathcal{B}(0), \cdots, \mathcal{B}(t)$, we have for any $0 \leq s \leq t$,
$$\begin{aligned} \left\|\frac{\partial L}{\partial W_i}(s)\right\|_F &\leq a_N \|W_{N:i+1}(s)\| \|\nabla L(a_N W_{N:1}(s))\|_F \|W_{i-1:1}(s)|_{\mathcal{R}(X)}\| \\ &\leq \frac{e^{2c_1}}{\sqrt{n_N}} \|\nabla L(a_N W_{N:1}(s))\|_F \\ &\leq \frac{e^{2c_1}}{\sqrt{n_N}}\sqrt{2\beta(1 - \eta\gamma)^s B}. \end{aligned} \tag{34}$$

Then,

$$\|W_i(t+1) - W_i(0)\|_F \leq \sum_{s=0}^{t} \|W_i(s+1) - W_i(s)\|_F$$

$$= \sum_{s=0}^{t} \left\| \eta \frac{\partial L}{\partial W_i}(s) \right\|_F$$

$$\leq \eta \frac{e^{2c_1}}{\sqrt{n_N}} \sqrt{2\beta B} \sum_{s=0}^{t} (1 - \eta\gamma)^{s/2}$$

$$\leq \eta \frac{e^{2c_1}}{\sqrt{n_N}} \sqrt{2\beta B} \sum_{s=0}^{t} (1 - \eta\gamma/2)^s$$

$$\leq \frac{2e^{2c_1}\sqrt{2\beta B}}{\sqrt{n_N}\gamma} = R.$$

This proves $\mathcal{C}(t+1)$. $\qquad\square$

*Proof of Claim 2.* Let $\delta_i = W_i(t) - W_i(0), 1 \leq i \leq N$. Using $\mathcal{C}(t)$, we have $\|\delta_i\|_F \leq R, 1 \leq i \leq N$. Set $\varepsilon_1 = e^{-c_1/2} \min\{e^{c_1} - e^{c_1/2}, e^{-c_2/2} - e^{-c_2}, 1/2\}$.
It is suffices to show that

$$\|W_{N:i}(t) - W_{N:i}(0)\| \leq e^{c_1/2}\varepsilon_1(n_{N-1}n_{N-1}\cdots n_{i-1})^{1/2}, 1 < i \leq N, \tag{35}$$

$$\left\|(W_{i:1}(t) - W_{i:1}(0))|_{\mathcal{R}(X)}\right\| \leq e^{c_1/2}\varepsilon_1(n_1 n_2 \cdots n_{i-1})^{1/2}, 1 \leq i < N, \tag{36}$$

and

$$\|W_{j:i}(t) - W_{j:i}(0)\| \leq M/2 \cdot N^\theta \left( \frac{1}{n_{min}} \prod_{i-1 \leq k \leq j} n_k \right)^{1/2}, 1 < i \leq j < N, \tag{37}$$

because $\sigma_{min}(A+B) \geq \sigma_{min}(A) - \sigma_{max}(B) = \sigma_{min}(A) - \|B\|$ and $\sigma_{max}(A+B) \leq \sigma_{max}(A) + \sigma_{max}(B) = \|A\| + \|B\|$ (e.g. see Theorem 1.3 in Chafaï et al. (2009)).
**Case 1.** We first prove (37).
For $1 \leq i < j \leq N$, we can write $W_{j:i}(t) = (W_j(0) + \delta_j) \cdots (W_i(0) + \delta_i)$.
Expanding the above product, each term has the form:

$$W_{j:(k_s+1)}(0) \cdot \delta_{k_s} \cdot W_{(k_s-1):(k_{s-1}+1)}(0) \cdot \delta_{k_{s-1}} \cdots \delta_{k_1} \cdot W_{(k_1-1):i}(0), \tag{38}$$

where $i \leq k_1 < \cdots < k_s \leq j$ are positions at which perturbation terms $\delta_{k_l}$ are taken out.
Notice that the convergence region assumption (31) implies that for any $1 < i \leq j < N$,

$$\|W_{j:i}(0)\| \leq M/2 \cdot N^\theta \left( \prod_{i \leq k \leq j-1} n_k \cdot \max\{n_{i-1}, n_j\} \right)^{1/2} \leq M \cdot N^\theta \left( \frac{\prod_{i-1 \leq k \leq j} n_k}{n_{min}} \right)^{1/2}. \tag{39}$$

WLOG, assume $M \geq 1$. If $i = j + 1$, then

$$\|W_{j:i}(0)\| = \|I\| \leq M \cdot N^\theta(n_j/n_{min})^{1/2}.$$

Assuming $i > 1, j < N$, and applying inequality (39) as well as the following inequality

$$\sum_{s=1}^{j-i+1} \binom{j-i+1}{s} x^s = (1+x)^{j-i+1} - 1 \leq (1+x)^N - 1, \forall x \geq 0,$$

we obtain

$$\|W_{j:i}(t) - W_{j:i}(0)\|$$

$$\leq \sum_{s=1}^{j-i+1} \binom{j-i+1}{s} R^s (M \cdot N^\theta)^{s+1} n_{min}^{-s/2}(n_{i-1} \cdots n_j/n_{min})^{1/2}$$

$$\leq M \cdot N^\theta(n_{i-1} \cdots n_j/n_{min})^{1/2}[(1 + R \cdot M \cdot N^\theta/\sqrt{n_{min}})^N - 1]$$

$$\leq \varepsilon_1 M \cdot N^\theta(n_{i-1} \cdots n_j/n_{min})^{1/2}.$$

The last line holds due to the following reasons: there exists absolute constant $A_1, A_2 > 0$ such that

$$(1 + x)^N - 1 \le A_2 x N,$$

if $x \ge 0$, $N \ge 1$, and $xN \le A_1$. Since there exists positive constant $C(c_1, c_2)$, which only depends on $c_1, c_2$, such that when

$$n_{min} \ge \frac{C(c_1, c_2) M^2 \kappa^2 B_0}{\varepsilon^2} N^{2\theta} n_N \tag{40}$$

we can have

$$R \cdot M \cdot N^{\theta+1} / \sqrt{n_{min}} \le A_1,$$

as well as

$$[(1 + R \cdot M \cdot N^\theta / \sqrt{n_{min}})^N - 1] \le A_2 \cdot M \cdot R \cdot N^{\theta+1} / \sqrt{n_{min}} \le \varepsilon_1 = \varepsilon_1(c_1, c_2).$$

**Case 2.** The proof of (35) is similar. Set $j = N$, we can save the factor $M \cdot N^\theta$ from previous calculation, which means

$$\|W_{N:i}(t) - W_{N:i}(0)\|$$

$$\le e^{c_1/2} \sum_{s=1}^{N-i+1} \binom{N-i+1}{s} R^s (M \cdot N^\theta)^s n_{min}^{-s/2} (n_{i-1} \cdots n_{N-1})^{1/2}$$

$$\le e^{c_1/2} (n_{i-1} \cdots n_{N-1})^{1/2} [(1 + R \cdot M \cdot N^\theta / \sqrt{n_{min}})^N - 1]$$

$$\le e^{c_1/2} \varepsilon_1 (n_{i-1} \cdots n_{N-1})^{1/2}, i \ge 2,$$

where the last line is implied by equation (40).

**Case 3.** Similarly, we have

$$\left\| W_{j:1}(t)|_{\mathcal{R}(X)} - W_{j:1}(0)|_{\mathcal{R}(X)} \right\|$$

$$\le e^{c_1/2} \sum_{s=1}^{j} \binom{j}{s} R^s (M \cdot N^\theta)^s n_{min}^{-s/2} (n_1 \cdots n_j)^{1/2}$$

$$\le e^{c_1/2} (n_1 \cdots n_j)^{1/2} [(1 + R \cdot M \cdot N^\theta / \sqrt{n_{min}})^N - 1]$$

$$\le e^{c_1/2} \varepsilon_1 (n_1 \cdots n_j)^{1/2}, j \le N - 1$$

This proves $\mathcal{B}(t)$.

$\square$

*Proof of Claim 3.* GD (7) implies

$$W_{N:1}(t+1)$$

$$= \left( W_N(t) - \eta \frac{\partial L^N}{\partial W_N}(t) \right) \left( W_{N-1}(t) - \eta \frac{\partial L^N}{\partial W_{N-1}}(t) \right) \cdots \left( W_1(t) - \eta \frac{\partial L^N}{\partial W_1}(t) \right)$$

$$= W_{N:1}(t) - \eta \cdot a_N \sum_{i=1}^{N} W_{N:i+1}(t) W_{N:i+1}^T(t) \nabla L(a_N W_{N:1}(t)) (W_{i-1:1}(t))^T (W_{i-1:1}(t)) + E(t),$$

where $E(t)$ contains all the high-order terms (those with $\eta^2$ or higher). We define a linear operator

$$P(t)[A] = a_N^2 \sum_{i=1}^{N} W_{N:i+1}(t) W_{N:i+1}^T(t) (A P_X) (W_{i-1:1}(t)|_{\mathcal{R}(X)})^T W_{i-1:1}(t)|_{\mathcal{R}(X)}, \tag{41}$$

for any $A \in \mathbb{R}^{n_N \times n_0}$.

Now we have

$$a_N W_{N:1}(t+1) = a_N W_{N:1}(t) - \eta \cdot P(t)[\nabla L(a_N W_{N:1}(t) P_X)] + a_N E(t). \tag{42}$$

Easy to check that $P(t)[\cdot]$ is a sum of positive semidefinite linear operator.

The following proposition describes the eigenvalues of the linear operator $P(t)[\cdot]$.

**Proposition D.2.** *Let $S_1$, $S_2$ be symmetric matrices. Suppose $S_1 = U\Lambda_1 U^T$, $S_2 = V\Lambda_2 V^T$, where $U = [u_1, u_2, \cdots, u_m]$, and $V = [v_1, v_2, \cdots, v_n]$ are orthogonal matrices, and $\Lambda_1 = diag(\lambda_1, \lambda_2, \cdots, \lambda_m)$ and $\Lambda_2 = diag(\mu_1, \mu_2, \cdots, \mu_n)$ are diagonal matrices. Then, the linear operator $L(A) := S_1 A S_2$ is orthogonally diagonalizable, and $L(A_{ij}) = \lambda_i \mu_j A_{ij}$, where $\lambda_i \mu_j$ represent all the eigenvalues corresponding to their eigenvectors $A_{ij} = u_i v_j^T$.*

Applying this proposition and the assumption $\mathcal{B}(t)$, we obtain the upper bound and lower bound for the maximum and minimum eigenvalues of the positive definite operator $P(t)$, respectively,

$$\lambda_{max}(P(t)) \leq a_N^2 \sum_{i=1}^{N} \sigma_{max}^2(W_{i-1:1}(t)|_{\mathcal{R}(X)}) \cdot \sigma_{max}^2(W_{N:i+1}(t)) \leq \frac{N}{n_N} e^{2c_1},$$

and

$$\lambda_{min}(P(t)) \geq a_N^2 \sum_{i=1}^{N} \sigma_{min}^2(W_{i-1:1}(t)|_{\mathcal{R}(X)}) \cdot \sigma_{min}^2(W_{N:i+1}(t)) \geq \frac{N}{n_N} e^{-2c_2}. \tag{43}$$

In conclusion, we have

$$\lambda_{max}(P(t)) \leq \frac{N}{n_N} e^{2c_1}, \text{ and } \lambda_{min}(P(t)) \geq \frac{N}{n_N} e^{-2c_2}. \tag{44}$$

With a learning rate of $\eta = \eta_\varepsilon = \frac{(1-\varepsilon)2n_N}{e^{6c_1+3c_2}\beta N}$, $0 < \varepsilon < 1$, we have

$$
\begin{aligned}
L_{t+1} &- L_t \\
&\leq \langle \nabla L_t, -\eta P(t)[\nabla L_t] \rangle_X + \langle \nabla L_t, a_N E(t) \rangle_X + \frac{\beta}{2} \|\eta P(t)[\nabla L_t] - a_N E(t)\|_X^2 \\
&= \langle \nabla L_t, -\eta P(t)[\nabla L_t] \rangle + \frac{\beta}{2}\eta^2 \|P(t)[\nabla L_t]\|_X^2 + F(t) \\
&\leq -\left(\eta \lambda_{min}(P(t)) - \frac{\beta}{2}\eta^2 \lambda_{max}^2(P(t))\right) \|\nabla L_t\|_X^2 + F(t) \\
&\leq -e^{-2c_2}\frac{N}{n_N}\eta \left(1 - e^{4c_1+2c_2}\frac{\beta}{2}\eta\frac{N}{n_N}\right) \|\nabla L_t\|_X^2 + F(t),
\end{aligned}
\tag{45}
$$

where

$$F(t) = \langle \nabla L_t, a_N E(t) \rangle_X + \frac{\beta}{2} \|\eta P(t)[\nabla L_t] - a_N E(t)\|_X^2 - \frac{\beta}{2}\eta^2 \|P(t)[\nabla L_t]\|_X^2.$$

We claim that $F(t)$ is sufficiently small, such that

$$
\begin{aligned}
L_{t+1} &- L_t \\
&\leq -e^{-2c_2}\frac{N}{n_N}\eta \left(1 - e^{4c_1+2c_2}\frac{\beta}{2}\eta\frac{N}{n_N}\right) \|\nabla L_t\|_X^2 + F(t) \\
&\leq -e^{-3c_2}\frac{N}{n_N}\eta \left(1 - e^{6c_1+3c_2}\frac{\beta}{2}\eta\frac{N}{n_N}\right) \|\nabla L_t\|_X^2 \\
&= -e^{-6(c_1+c_2)}\frac{2\varepsilon(1-\varepsilon)}{\beta} \|\nabla L_t\|_X^2.
\end{aligned}
\tag{46}
$$

Assuming this claim for the moment, we complete the proof. Combining (26) and (46), we have

$$
\begin{cases}
L_{t+1} - L_t \leq -e^{-6(c_1+c_2)}\frac{2\varepsilon(1-\varepsilon)}{\beta} \|\nabla L_t\|_X^2, \\
L(W_*) - L_t \geq -\frac{1}{2\alpha} \|\nabla L_t\|_X^2,
\end{cases}
$$

which implies

$$L_{t+1} - L(W_*) \leq \left(1 - e^{-6(c_1+c_2)}\frac{4\varepsilon(1-\varepsilon)}{\kappa}\right)(L_t - L(W_*)), \tag{47}$$

that is

$$L_t - L(W_*) \leq \left(1 - e^{-6(c_1+c_2)} \frac{4\varepsilon(1-\varepsilon)}{\kappa}\right)^t (L_0 - L(W_*)) = (1 - \eta\gamma)^t (L_0 - L(W_*)). \quad (48)$$

While estimating $F(t)$, we observe that

$$|F(t)|$$
$$\leq \|\nabla L_t\|_X \|a_N E(t)\|_X + \frac{\beta}{2}(2\eta\lambda_{max}(P(t)) \|\nabla L_t\|_X \|a_N E(t)\|_X + \|a_N E(t)\|_X^2)$$
$$=: I_1 + I_2.$$

From (34), we have

$$\left\|\frac{\partial L}{\partial W_i}(t)\right\|_F \leq \frac{e^{2c_1}}{\sqrt{n_N}} \|\nabla L(a_N W_{N:1}(t))\|_F = \frac{e^{2c_1}}{\sqrt{n_N}} \|\nabla L(a_N W_{N:1}(t))\|_X =: K.$$

Expanding the product

$$W_{N:1}(t+1) = \left(W_N(t) - \eta\frac{\partial L^N}{\partial W_N}(t)\right)\left(W_{N-1}(t) - \eta\frac{\partial L^N}{\partial W_{N-1}}(t)\right)\cdots\left(W_1(t) - \eta\frac{\partial L^N}{\partial W_1}(t)\right),$$

each term has the form:

$$\Delta = W_{N:(k_s+1)}(t) \cdot \eta\frac{\partial L}{\partial W_{k_s}}(t) \cdot W_{(k_s-1):(k_{s-1}+1)}(t) \cdot \eta\frac{\partial L}{\partial W_{k_{s-1}}}(t) \cdots \eta\frac{\partial L}{W_{k_1}}(t) \cdot W_{(k_1-1):1}(t),$$

where $1 \leq k_1 < k_2 < \cdots < k_s \leq N$.

As a direct consequence of inequality $\mathcal{B}(t)$ and inequality (39), we obtain

$$\|\Delta\|_X = \|\Delta P_X\|_F \leq \frac{1}{a_N \sqrt{n_N}} e^{2c_1}(\eta K)^s \left(\frac{M \cdot N^\theta}{\sqrt{n_{min}}}\right)^{s-1},$$

Recall that $E(t)$ contains all high-order terms (those with $\eta^2$ or higher) in the expansion of the product. Thus, $E(t)$ can be expressed as follows:

$$\sum_{s=2}^{N} \sum_{1 \leq k_1 < k_2 < \cdots < k_s \leq N} W_{N:(k_s+1)}(t) \cdot \eta\frac{\partial L}{\partial W_{k_s}}(t) \cdot W_{(k_s-1):(k_{s-1}+1)}(t) \cdot \eta\frac{\partial L}{\partial W_{k_{s-1}}}(t) \cdots \eta\frac{\partial L}{W_{k_1}}(t) \cdot W_{(k_1-1):1}(t).$$

Set $\xi = \min\{(e^{-2c_2} - e^{-3c_2})/e^{4c_1+1}, \frac{1}{4}(e^{6c_1} - e^{4c_1})/e^{6c_1+1}, \frac{1}{2}(e^{6c_1} - e^{4c_1})^{1/2}/e^{4c_1+1}, 1\}$.

Recall the inequality $\binom{N}{s} \leq (eN)^s$. Thus, we have

$$a_N \|E(t)\|_X$$
$$\leq \frac{1}{\sqrt{n_N}} e^{2c_1} \sum_{s=2}^{N} \binom{N}{s} (\eta K)^s \left(\frac{M \cdot N^\theta}{\sqrt{n_{min}}}\right)^{s-1}$$
$$\leq \frac{1}{\sqrt{n_N}} \left(\frac{M \cdot N^\theta}{\sqrt{n_{min}}}\right)^{-1} e^{2c_1} \sum_{s=2}^{N} (eN)^s (\eta K)^s \left(\frac{M \cdot N^\theta}{\sqrt{n_{min}}}\right)^{s}$$
$$\leq \frac{1}{\sqrt{n_N}} e^{2c_1} (\eta eKN) \frac{\eta eKM \cdot N^{\theta+1}/\sqrt{n_{min}}}{1 - \eta eKM \cdot N^{\theta+1}/\sqrt{n_{min}}}$$
$$\leq \xi \frac{N}{n_N} \eta \cdot e^{4c_1+1} \|\nabla L(a_N W_{N:1}(t))\|_X \; (\text{if } \eta eKM \cdot N^{\theta+1}/\sqrt{n_{min}} < \xi/(1+\xi))$$
$$= \xi \cdot e^{4c_1+1} \left(\eta\frac{N}{n_N}\right) \|\nabla L(a_N W_{N:1}(t))\|_X. \quad (49)$$

Using (33) and the upper bound of $\eta$, we know that there exists constant $C(c_1, c_2)$, such that

$$n_{min} \geq C(c_1, c_2)M^2 \cdot B_0 N^{2\theta} n_N,$$

and

$$\eta e K M \cdot N^{\theta+1}/\sqrt{n_{min}} \leq \frac{2\sqrt{2}M \cdot e^{1+2c_1}\sqrt{B_0}N^\theta\sqrt{n_N}}{\sqrt{n_{min}}} = \frac{1}{C'(c_1,c_2)} \leq \frac{\xi}{2} \leq \frac{\xi}{1+\xi}.$$

Using (49), we have

$$I_1 \leq \xi \cdot e^{4c_1+1}\left(\eta\frac{N}{n_N}\right)\|\nabla L_t\|_X^2 \leq (e^{-2c_2} - e^{-3c_2})\left(\eta\frac{N}{n_N}\right)\|\nabla L_t\|_X^2, \tag{50}$$

and

$$\begin{aligned}
&I_2\\
\leq& \frac{\beta}{2}\left(2\xi \cdot e^{6c_1+1}\left(\eta^2\frac{N^2}{n_N^2}\right)\|\nabla L_t\|_X^2 + \xi^2 \cdot e^{8c_1+2}\left(\eta^2\frac{N^2}{n_N^2}\right)\|\nabla L_t\|_X^2\right)\\
\leq& (e^{6c_1} - e^{4c_1})\frac{\beta}{2}\eta^2\frac{N^2}{n_N^2}\|\nabla L_t\|_X^2.
\end{aligned}$$

Thus, (46) valid.
This proves $\mathcal{A}(t)$.

$\square$

As a direct consequence of the proof of Lemma D.1, we can obtain the following lemma.

**Lemma D.3.** *Assume all assumptions in Lemma D.1 hold. For any $\tau > 0$, we can choose new constants $c_1, c_2$ as well as $C := C(c_1, c_2)$ such that the overparameterization assumption (32) in Lemma D.1 hold and*

$$\|R(t)\|_X \leq \tau \|a_N W_{N:1}(t) - W_*\|_X, \tag{51}$$

*where*

$$a_N W_{N:1}(t+1) = a_N W_{N:1}(t) - \frac{N}{n_N}\eta\nabla L(a_N W_{N:1}(t)) + R(t).$$

*Proof of Lemma D.3.* Due to (33), (42), (44), (49), and lemma C.2, we have

$$\begin{aligned}
\|R(t)\|_X &= \left\|a_N E(t) + \eta\left(\frac{N}{n_N}\nabla L_t - P(t)[\nabla L_t]\right)\right\|_X\\
&\leq \|a_N E(t)\|_X + \eta\max\left\{\lambda_{max}(P(t)) - \frac{N}{n_N}, \frac{N}{n_N} - \lambda_{min}(P(t))\right\}\|\nabla L_t\|_X\\
&\leq (C' \cdot \xi + \max\{e^{2c_1} - 1, 1 - e^{-2c_2}\}) \cdot \eta\frac{N}{n_N} \cdot \|\nabla L_t\|_X\\
&\leq \frac{2\sqrt{2\beta(L_t - L(W_*))}}{e^{6c_1+3c_2} \cdot \beta} \cdot (C' \cdot \xi + \max\{e^{2c_1} - 1, 1 - e^{-2c_2}\}).
\end{aligned}$$

Because $L_t - L(W_*)$ is non-increasing in $t$, and $C'$ is a constant that depends only on $c_1, c_2$, we can choose a sufficiently small positive $c_1, c_2$ and $\xi$, which depends on $\tau$, such that

$$\|R(t)\|_X \leq \tau\frac{\sqrt{2\beta(L_t - L(W_*))}}{\beta} \leq \tau\|a_N W_{N:1}(t) - W_*\|_X.$$

$\square$

**Lemma D.4.** *Assume $\tau \in [0, 1)$. Consider a discrete dynamical system $V(t)$ such that,*

$$V(t+1) = V(t) - \eta_*\nabla L(V(t)) + R(t),$$

*where $\|R(t)\|_X \leq \tau\|V(t) - W_*\|_X$. If $\eta_* \leq 2/\beta$, we have*

$$\|V(t) - W_*\|_X^2 \leq (q + 7\tau)^t\|V(0) - W_*\|_X^2,$$

*where $q$ is defined in (15).*

*Proof of Lemma D.4.* Set $\Delta(t) = V(t) - W_*$ and $\tau' = \tau \|\Delta(t)\|_X$. Notice that

$$\Delta(t+1) = \Delta(t) - \eta_*(\nabla L(V(t)) - \nabla L(W_*)) + R(t),$$

and

$$\begin{aligned}
&\|\Delta(t+1)\|_X^2\\
&\leq \eta_*^2 \|\nabla L(V(t)) - \nabla L(W_*)\|_X^2 - 2\eta_* \langle \Delta(t), \nabla L(V(t)) - \nabla L(W_*) \rangle_X\\
&\quad + \|\Delta(t)\|_X^2 + (2\|\Delta(t)\|_X + 2\eta_* \|\nabla L(V(t)) - \nabla L(W_*)\|_X + \tau')\tau'.
\end{aligned}$$

By inequality (23),

$$\begin{aligned}
&\|\Delta(t+1)\|_X^2\\
&\leq \|\Delta(t)\|_X^2 - 2\eta_* \langle \Delta(t), \nabla L(V(t)) - \nabla L(W_*) \rangle_X\\
&\quad + \eta_*^2 \|\nabla L(V(t)) - \nabla L(W_*)\|_X^2 + 7\tau \|\Delta(t)\|_X^2\\
&= (1 + 7\tau) \|\Delta(t)\|_X^2 - 2\eta_* \langle \Delta(t), \nabla L(V(t)) - \nabla L(W_*) \rangle_X\\
&\quad + \eta_*^2 \|\nabla L(V(t)) - \nabla L(W_*)\|_X^2\\
&\leq (1 + 7\tau) \|\Delta(t)\|_X^2 - 2\eta_* \frac{\alpha\beta}{\alpha+\beta} \|\Delta(t)\|_X^2\\
&\quad + \left(\eta_*^2 - \frac{2\eta_*}{\alpha+\beta}\right) \|\nabla L(V(t)) - \nabla L(W_*)\|_X^2.
\end{aligned}$$

**Case 1**: $\frac{2}{\alpha+\beta} < \eta_* < \frac{2}{\beta}$.
In this case, we have

$$\begin{aligned}
&\|\Delta(t+1)\|_X^2\\
&\leq (1+7\tau) \|\Delta(t)\|_X^2 - 2\eta_* \frac{\alpha\beta}{\alpha+\beta} \|\Delta(t)\|_X^2 + \left(\eta_*^2 - \frac{2\eta_*}{\alpha+\beta}\right) \|\nabla L(V(t)) - \nabla L(W_*)\|_X^2\\
&\leq (1+7\tau) \|\Delta(t)\|_X^2 - 2\eta_* \frac{\alpha\beta}{\alpha+\beta} \|\Delta(t)\|_X^2 + \left(\eta_*^2 - \frac{2\eta_*}{\alpha+\beta}\right) \beta^2 \|\Delta(t)\|_X^2\\
&\leq (1 + 7\tau - \beta\eta_*(2 - \eta_*\beta)) \|\Delta(t)\|_X^2\\
&= (q + 7\tau) \|\Delta(t)\|_X^2.
\end{aligned}$$

**Case 2**: $0 < \eta_* \leq \frac{2}{\alpha+\beta}$.
Similarly, we have

$$\|\Delta(t+1)\|_X^2 \leq (1 + 7\tau - \alpha\eta_*(2 - \eta_*\alpha)) \|\Delta(t)\|_X^2 = (q + 7\tau) \|\Delta(t)\|_X^2.$$

In both cases, we have $\|\Delta(t+1)\|_X^2 \leq (q + 7\tau) \|\Delta(t)\|_X^2$.

Thus, $\|\Delta(t)\|_X^2 \leq (q + 7\tau)^t \|\Delta(0)\|_X^2$.

$\square$

Next, we will show that the trajectories of the GD (30) for deep linear neural networks (29) are close to those of GD (2) for the corresponding convex problem (1).

**Lemma D.5.** *Consider the GD for the deep linear neural networks (30) with learning rate $\eta < \eta_1$ for $a_N W_{N:1}(t), t = 0, 1, \cdots$, and the GD (2) with learning rate $\eta_* = \frac{N}{n_N}\eta$ for $W(t), t = 0, 1, \cdots$.*

*Assume $C(c_1, c_2)$ exists in Lemma D.1 for any $c_1, c_2 > 0$. For any $\tau \in (0, 1)$, $\eta < \eta_1$ ($\eta_1$ defined in B), we can choose $c_1, c_2 > 0$ and the constant $C = C(c_1, c_2) = C'(\tau, \eta/\eta_1)$, such that inequality (51) holds, given initialization condition (31), and overparameterization condition*

$$n_{min} \geq CM^2 \kappa^2 B_0 N^{2\theta} n_N. \tag{52}$$

*Furthermore, we have*

$$\|a_N W_{N:1}(t) - W(t)\|_X^2 \le D(\tau, q, t) \|a_N W_{N:1}(0) - W_*\|_X^2, \tag{53a}$$

$$|\mathcal{E}_{DLN}(t) - \mathcal{E}(t)| \le \beta \left( q^{t/2} \sqrt{D(\tau, q, t)} + \frac{1}{2} D(\tau, q, t) \right) \|a_N W_{N:1}(0) - W_*\|_X^2, \tag{53b}$$

$$\mathcal{E}_{DLN}(t) \le 3\beta(q + \tau)^t \|a_N W_{N:1}(0) - W_*\|_X^2, \tag{53c}$$

*where $D(\tau, q, t) = \min\left\{ \frac{\tau}{1-q}, 2(q+\tau)^t \right\}$, with $q$ defined in (15).*

*Proof of Lemma D.5.* Using Lemma D.3, we obtain that for any $\tau \in (0, 1)$ and $\eta < \eta_1$, we can find sufficiently small positive constants $c_1, c_2$, which only depend on $\tau, \eta/\eta_1$, and constant $C = C(c_1, c_2) = C''(\tau, \eta/\eta_1)$ mentioned in Lemma D.3, such that

$$\eta = \frac{(1-\varepsilon)2n_N}{e^{6c_1+3c_2}\beta N},$$

where $0 < \varepsilon < 1$, as well as

$$V(t+1) = V(t) - \eta_* \nabla L(V(t)) + R(t),$$

where $V(t) = a_N W_{N:1}(t)$, $\eta_* = \frac{N}{n_N}\eta$, and $\|R(t)\|_X \le \tau' = \tau \|V(t) - W_*\|_X$.

Notice that $\theta_0 := \eta/\eta_1 = \frac{1-\varepsilon}{e^{6c_1+3c_2}}$ and $\eta/\eta_0 = 1 - \varepsilon$, where $\eta_0 = \frac{2n_N}{e^{6c_1+3c_2}\beta N}$.

For the right hand side of inequality (32), we have

$$\frac{C(c_1, c_2)M^2\kappa^2 B_0}{\varepsilon^2} N^{2\theta} n_N = \frac{C''(\tau, \eta/\eta_1)M^2\kappa^2 B_0}{\varepsilon^2} N^{2\theta} n_N.$$

To show that inequality (32) is equivalent to inequality (52), it suffices to show that $\varepsilon$ only depend on $\tau, \eta/\eta_1$. Notice that

$$\varepsilon = 1 - \eta/\eta_0 = 1 - \theta_0 e^{6c_1+3c_2},$$

and $c_1, c_2$ only depend on $\tau$ and $\eta/\eta_1$, which implies $\varepsilon$ only depend on $\tau, \eta/\eta_1$.

Now, we will prove the three inequalities in (53).

Recall GD (2) for $W(t)$. Define $\Delta(t) = V(t) - W(t) = a_N W_{N:1}(t) - W(t)$. Notice that

$$\Delta(t+1) = \Delta(t) - \eta_*(\nabla L(V(t)) - \nabla L(W(t))) + R(t),$$

and

$$\begin{aligned}
&\|\Delta(t+1)\|_X^2 \\
\le &\eta_*^2 \|\nabla L(V(t)) - \nabla L(W(t))\|_X^2 - 2\eta_* \langle \Delta(t), \nabla L(V(t)) - \nabla L(W(t)) \rangle_X \\
&+ \|\Delta(t)\|_X^2 + (2\|\Delta(t)\|_X + 2\eta_* \|\nabla L(V(t)) - \nabla L(W(t))\|_X + \tau')\tau'.
\end{aligned}$$

Let $l_t = 2\|\Delta(t)\|_X + 2\eta_* \|\nabla L(V(t)) - \nabla L(W(t))\|_X + \tau'$.

Now, we aim to find an upper bound for $l_t$.

Applying lemma C.2 with the assumption $0 < \eta_* = \frac{N}{n_N}\eta < \frac{2}{\beta}$, we know that

$$l_t \le (6\|\Delta(t)\|_X + \tau') \le 7(\|W(t) - W_*\|_X + \|V(t) - W_*\|_X). \tag{54}$$

Thus

$$l_t \tau' \le 7\tau \|V(t) - W_*\|_X (\|V(t) - W_*\|_X + \|W(t) - W_*\|_X) =: U_t \tau.$$

By inequality (23),

$$\|\Delta(t+1)\|_X^2$$
$$\leq \|\Delta(t)\|_X^2 - 2\eta_* \langle \Delta(t), \nabla L(V(t)) - \nabla L(W(t)) \rangle_X$$
$$+ \eta_*^2 \|\nabla L(V(t)) - \nabla L(W(t))\|_X^2 + U_t \tau$$
$$= \|\Delta(t)\|_X^2 - 2\eta_* \langle V(t) - W(t), \nabla L(V(t)) - \nabla L(W(t)) \rangle_X$$
$$+ \eta_*^2 \|\nabla L(V(t)) - \nabla L(W(t))\|_X^2 + U_t \tau$$
$$\leq \|\Delta(t)\|_X^2 - 2\eta_* \frac{\alpha\beta}{\alpha+\beta} \|\Delta(t)\|_X^2$$
$$+ \left( \eta_*^2 - \frac{2\eta_*}{\alpha+\beta} \right) \|\nabla L(V(t)) - \nabla L(W(t))\|_X^2 + U_t \tau.$$

**Case 1**: $\frac{2}{\alpha+\beta} < \eta_* < \frac{2}{\beta}$.
In this case, we have

$$\|\Delta(t+1)\|_X^2$$
$$\leq \|\Delta(t)\|_X^2 - 2\eta_* \frac{\alpha\beta}{\alpha+\beta} \|\Delta(t)\|_X^2 + \left( \eta_*^2 - \frac{2\eta_*}{\alpha+\beta} \right) \|\nabla L(V(t)) - \nabla L(W(t))\|_X^2 + U_t \tau$$
$$\leq \|\Delta(t)\|_X^2 - 2\eta_* \frac{\alpha\beta}{\alpha+\beta} \|\Delta(t)\|_X^2 + \left( \eta_*^2 - \frac{2\eta_*}{\alpha+\beta} \right) \beta^2 \|\Delta(t)\|_X^2 + U_t \tau$$
$$\leq (1 - \beta\eta_*(2 - \eta_*\beta)) \|\Delta(t)\|_X^2 + U_t \tau$$
$$=: q \|\Delta(t)\|_X^2 + U_t \tau.$$

**Case 2**: $0 < \eta_* \leq \frac{2}{\alpha+\beta}$.
Similarly, we have

$$\|\Delta(t+1)\|_X^2 \leq (1 - \alpha\eta_*(2 - \eta_*\alpha)) \|\Delta(t)\|_X^2 + U_t \tau =: q \|\Delta(t)\|_X^2 + U_t \tau. \tag{55}$$

In both cases, we have $0 < q < 1$.

First of all, since $U_t \leq U_0$ and $\|\Delta(0)\|_X = 0$, we obtain that

$$\|\Delta(t)\|_X^2 \leq \frac{U_0\tau}{1-q} + q^t \left( \|\Delta(0)\|_X^2 - \frac{U_0\tau}{1-q} \right) \leq \frac{U_0\tau}{1-q} \leq \frac{14\tau}{1-q} \|V(0) - W_*\|_X^2.$$

Applying Lemma D.4 for $V(t)$ and $W(t)$, we obtain $\|V(t) - W_*\|_X^2 \leq (1+\varepsilon)^t q^t \|V(0) - W_*\|_X^2$ and $\|W(t) - W_*\|_X^2 \leq q^t \|W(0) - W_*\|_X^2$, respectively. Thus,

$$|L(W(t)) - L(a_N W_{N:1}(t))|$$
$$\leq |\langle \nabla L(W(t)), \Delta(t) \rangle_X| + \frac{\beta}{2} \|\Delta(t)\|_X^2$$
$$\leq \beta \|W(t) - W_*\|_X \cdot \|\Delta(t)\|_X + \frac{\beta}{2} \|\Delta(t)\|_X^2$$
$$\leq \beta \left( q^{t/2} \sqrt{\frac{14\tau}{1-q}} + \frac{7\tau}{1-q} \right) \|V(0) - W_*\|_X^2.$$

Generally speaking, (55) implies

$$\|\Delta(t)\|_X^2 \leq \tau \sum_{j=0}^{t-1} q^{t-1-j} U_j.$$

We have

$$\|\Delta(t)\|_X^2 \leq 14\tau \sum_{j=0}^{t-1} (q + 7\tau)^j q^{t-1-j} \|V(0) - W_*\|_X^2$$
$$\leq 2(q + 7\tau)^t \left( 1 - \left(\frac{q}{q+7\tau}\right)^t \right) \|V(0) - W_*\|_X^2$$

Thus, we have

$$\|a_N W_{N:1}(t) - W(t)\|_X^2 \leq \min\left\{\frac{14\tau}{1-q}, 2(q+7\tau)^t\right\} \|V(0) - W_*\|_X^2 \,,$$

as well as

$$|L(W(t)) - L(a_N W_{N:1}(t))|$$

$$\leq \beta \|W(t) - W_*\|_X \cdot \|\Delta(t)\|_X + \frac{\beta}{2} \|\Delta(t)\|_X^2$$

$$\leq \beta \left(\sqrt{\min\left\{\frac{14\tau}{1-q}, 2(q+7\tau)^t\right\}} \cdot q^{t/2} + \frac{1}{2}\min\left\{\frac{14\tau}{1-q}, 2(q+7\tau)^t\right\}\right) \|V(0) - W_*\|_X^2 \,.$$

By triangle inequality as well as $L(W(t)) - L(W_*) \leq \frac{\beta}{2}q^t \|V(0) - W_*\|_X^2$, we have

$$|L(a_N W_{N:1}(t)) - L(W_*)| \leq 3\beta(q+7\tau)^t \|V(0) - W_*\|_X^2 \,.$$

Without loss of generality, we replace all $14\tau$ and $7\tau$ by $\tau$, which completes the proof.

$\square$

## E  GAUSSIAN INITIALIZATION FALL INTO THE CONVERGENCE REGION

In this section, we first establish some spectral properties of the products of random Gaussian matrices. The spectral properties lead to the conclusion: overparameterization guarantees that the random initialization will fall into the convergence region with high probability.

**Gaussian initialization**:

Denote by $N(0,1)$ the standard Gaussian distribution, and $\chi_k^2$ the chi square distribution with $k$ degrees of freedom. Let $S^{d-1} = \{x \in \mathbb{R}^d; \|x\|_2 = 1\}$ be the unit sphere in $\mathbb{R}^d$.

The scaling factor $a_N = \frac{1}{\sqrt{n_1 n_2 \cdots n_N}}$ ensures that the networks at initialization preserves the norm of every input in expectation.

**Lemma E.1.** *For any $x \in \mathbb{R}^{n_0}$, the Gaussian initialization satisfies*

$$\mathbb{E}\left[\|a_N W_{N:1}(0)x\|_2^2\right] = \|x\|_2^2 \,.$$

*Proof of Lemma E.1.* For random matrix $A \in \mathbb{R}^{n_i \times n_{i-1}}$ with i.i.d $N(0,1)$ entries and any vector $0 \neq v \in \mathbb{R}^{n_{i-1}}$, the distribution of $\frac{\|Av\|_2^2}{\|v\|_2^2}$ is $\chi_{n_i^2}$. We rewrite

$$\|W_{N:1}(0)x\|_2^2 \,/\, \|x\|_2^2 = Z_N Z_{N-1} \cdots Z_1,$$

where $Z_i = \|W_{i:1}(0)x\|^2 \,/\, \|W_{i-1:1}(0)x\|^2$.

Then we know that the distribution of random variable $Z_1 \sim \chi_{n_1}^2$, and conditional distribution of random variables $Z_i | (Z_1, \cdots, Z_{i-1}) \sim \chi_{n_i}^2 (1 < i \leq N)$. Thus, $Z_1, \cdots, Z_{n_i}$ are independent. By the law of iterated expectations, we have

$$\mathbb{E}[\|W_{N:1}(0)x\|_2^2 / \|x\|_2^2] = \prod_{j=1}^{N} n_j.$$

$\square$

Define $\Delta_1 = \sum_{j=1}^{N-1} 1/n_j$. Now, we introduce a new notation $\Omega\left(\frac{1}{\Delta_1}\right)$, which means that there exists $k > 0$, such that $\Omega\left(\frac{1}{\Delta_1}\right) \geq \frac{k}{\Delta_1}$.

**Lemma E.2.** *Consider real random matrix $A_j \in \mathbb{R}^{n_j \times n_{j-1}}, 1 \leq j \leq q$ with i.i.d $N(0,1)$ entries and any vector $0 \neq x \in \mathbb{R}^{n_1}$.*
*Define $\Delta_1(q) = \sum_{j=1}^{q} \frac{1}{n_j}$ and $n_{min} = \min_{1 \leq j \leq q} n_j$. Then*

$$\mathbb{P}(\|A_q A_{q-1} \cdots A_1 x\|_2^2 / \|x\|_2^2 > e^c n_1 \cdots n_q) \leq \exp\left\{-\frac{c^2}{8\Delta_1(q)}\right\} =: f_1(c), \forall c > 0. \quad (56)$$

*When $0 < c \le 3\ln 2$, $\Delta_1(q) \le c/(12\ln 2)$, we have*

$$\mathbb{P}(\|A_q A_{q-1}\cdots A_1 x\|_2^2 / \|x\|_2^2 < e^{-c} n_1 \cdots n_q) \le \exp\left\{-\frac{c^2}{36\ln(2)\Delta_1(q)}\right\} =: f_2(c). \quad (57)$$

*Hence, for any $x \in S^{n_0-1}$ with probability at least $1 - e^{-\Omega(\frac{1}{\Delta_1(q)})}$, we have*

$$e^{-c_2/2}(n_1\cdots n_q)^{1/2} \le \|A_q\cdots A_1 x\|_2 \le e^{c_1/2}(n_1\cdots n_q)^{1/2},$$

*when $0 < c_2 \le 3\ln 2$, $\Delta_1(q) \le c_2/(12\ln 2)$.*

*Proof of Lemma E.2.* For random matrix $A_i \in \mathbb{R}^{n_i \times n_{i-1}}$ with i.i.d $N(0,1)$ entries and any vector $0 \ne v \in \mathbb{R}^{n_{i-1}}$, the random variable $\frac{\|A_i v\|_2^2}{\|v\|_2^2}$ is distributed as $\chi_{n_i}^2$. We rewrite

$$\|A_q\cdots A_1 x\|_2^2 / \|x\|_2^2 = Z_q Z_{q-1}\cdots Z_1,$$

where $Z_i = \|A_{i:1}x\|^2 / \|A_{i-1:1}x\|^2$. We have $Z_1 \sim \chi_{n_1}^2$, $Z_i|(Z_1,\cdots,Z_{i-1}) \sim \chi_{n_i}^2 (1 < i \le q)$. Recall the moments of $Z \sim \chi_m^2$:

$$\mathbb{E}[Z^\lambda] = \frac{2^\lambda \Gamma(\frac{m}{2}+\lambda)}{\Gamma(\frac{m}{2})}, \forall \lambda > -\frac{m}{2}.$$

Now, we aim to find the Chernoff type bound.

**Case 1:** We define ratio of Gamma function

$$R(x,\lambda) = \frac{\Gamma(x+\lambda)}{\Gamma(x)}, \lambda > 0, x > 0.$$

In Jameson (2013), we have

$$R(x,\lambda) \le x(x+\lambda)^{\lambda-1} \le (x+\lambda)^\lambda, \lambda > 0, x > 0. \quad (58)$$

Fixed $c > 0$, for any $\lambda > 0$ we have

$$
\begin{aligned}
\mathbb{P}(Z_q\cdots Z_1 > e^c n_1\cdots n_q) &\le \mathbb{P}((Z_q\cdots Z_1)^\lambda > e^{\lambda c}(n_1\cdots n_q)^\lambda) \\
&\le e^{-\lambda c}(n_1\cdots n_q)^{-\lambda}\mathbb{E}[(Z_q\cdots Z_1)^\lambda] \quad \text{(Markov inequality)} \\
&= \exp\{-\lambda(c + \ln(n_1\cdots n_q))\}\prod_{j=1}^q 2^\lambda R(n_j/2,\lambda) \quad \text{(Law of total expectation)} \\
&\le \exp\{-\lambda(c + \ln(n_1\cdots n_q)) + q\lambda\ln 2 + \sum_{j=1}^q \lambda\ln(\frac{n_j}{2}+\lambda)\} \text{(Inequality (58))} \\
&= \exp\{-\lambda c + \lambda\sum_{j=1}^q \ln(1+\frac{2\lambda}{n_j})\} \\
&\le \exp\{-\lambda c + 2\lambda^2\sum_{j=1}^q \frac{1}{n_j}\}.
\end{aligned}
$$

Define constant $\Delta_1(q) = \sum_{j=1}^q \frac{1}{n_j}$. Set $\lambda = \frac{c}{4\Delta_1(q)}$, we obtain (56).

**Case 2:** Let $n_{min} = \min_{1\le j\le q} n_j$.

$$
\begin{aligned}
\mathbb{P}(Z_q\cdots Z_1 < e^{-c} n_1\cdots n_q) &\le \mathbb{P}((Z_q\cdots Z_1)^\lambda > e^{-\lambda c}(n_1\cdots n_q)^\lambda) \\
&\le \exp\{\lambda(c - \ln(n_1\cdots n_q)) + q\lambda\ln 2 + \sum_{j=1}^q \ln R(\frac{n_j}{2},\lambda)\}.
\end{aligned}
$$

Define

$$f(\lambda) = \lambda(c - \ln(n_1 \cdots n_q)) + q\lambda \ln 2 + \sum_{j=1}^{q} \ln R(\frac{n_j}{2}, \lambda), -\frac{n_{min}}{2} < \lambda \le 0.$$

Notice that $f(0) = 0$. Define digamma function,

$$\psi(x) = \frac{d}{dx} \ln(\Gamma(x)) = \frac{\Gamma'(x)}{\Gamma(x)}.$$

Qi et al. (2006) proved the following sharp inequality of digamma function,

$$\ln(x + \frac{1}{2}) - \frac{1}{x} < \psi(x) < \ln(x + e^{-\gamma}) - \frac{1}{x}, x > 0,$$

where $\gamma$ is the Euler-Mascheroni constant, and $e^{-\gamma} \approx 0.561459$.
Thus,

$$f'(\lambda) = c + \sum_{j=1}^{q} \left[ -\ln(\frac{n_j}{2}) + \psi(\frac{n_j}{2} + \lambda) \right] \ge c + \sum_{j=1}^{q} \ln(1 + \frac{\lambda + 1/2}{n_j/2}) - \sum_{j=1}^{q} \frac{1}{n_j/2 + \lambda}.$$

Since $\ln(1 + x)$ is concave, we have

$$\ln(1 + x) \ge 2\ln(2)x, x \in [-1/2, 0].$$

If $-\frac{n_{min}}{4} \le \lambda \le 0$, then

$$f(\lambda) = f(0) - \int_{\lambda}^{0} f'(x)dx$$

$$\le c\lambda + \int_{0}^{\lambda} \left[ \sum_{j=1}^{q} \ln(1 + \frac{x + 1/2}{n_j/2}) - \sum_{j=1}^{q} \frac{1}{n_j/2 + x} \right] dx$$

$$= c\lambda + \sum_{j=1}^{q} \left[ \lambda \ln(1 + \frac{\lambda + 1/2}{n_j/2}) + (n_j/2 + 1/2)\ln(1 + \frac{\lambda}{n_j/2 + 1/2}) - \lambda - \ln(1 + \frac{\lambda}{n_j/2}) \right]$$

$$\le c\lambda + \sum_{j=1}^{q} (\lambda - 1)\ln(1 + \frac{\lambda}{n_j/2})$$

$$\le c\lambda + 4\ln(2)\lambda(\lambda - 1)\Delta_1(q).$$

Assume $0 < c \le 3\ln 2$. Let $A = 12\ln 2$, and $\lambda^* = -\frac{c}{A\Delta_1(q)}$. Since $n_{min}\Delta_1(q) \ge 1$, we have $\lambda^* \ge -n_{min}/4$.
Assume $\Delta_1(q) \le c/(12\ln 2)$.
Thus

$$f(\lambda^*) \le -\frac{c^2}{A\Delta_1(q)} + 4\ln 2 \frac{c^2}{A\Delta_1(q)} \left( \frac{\Delta_1(q)}{c} + \frac{1}{A} \right) \le -\frac{c^2}{36\ln(2)\Delta_1(q)}. \tag{59}$$

Thus, we obtain (57). $\qquad\square$

**Lemma E.3.** *There exists a positive constant $C(c_1, c_2)$ which only depends on $c_1, c_2$, such that if $n_N\Delta_1 \le C(c_1, c_2)$, then for any fixed $1 < i \le N$, with probability at least $1 - \exp\left\{ -\Omega\left( \frac{1}{\Delta_1} \right) \right\}$ we have*

$$\sigma_{\max}(W_{N:i}(0)) \le e^{c_1}(n_{i-1}n_i \cdots n_{N-1})^{1/2}, \tag{60}$$

*and*

$$\sigma_{\min}(W_{N:i}(0)) \ge e^{-c_2}(n_{i-1}n_i \cdots n_{N-1})^{1/2}. \tag{61}$$

*Proof of Lemma E.3.* Let $A = W_{N:i}^T(0)$. We know that

$$\sigma_{max}(A) = \|A\| = \sup_{v \in S^{n_N - 1}} \|Av\|_2$$

and

$$\sigma_{min}(A) = \inf_{v \in S^{n_N - 1}} \|Av\|_2 .$$

Applying lemma E.2, we know that with probability at least $1 - \exp\left\{-\Omega\left(\frac{1}{\Delta_1}\right)\right\}$,

$$\|Av\|_2 / \|v\|_2 \in [e^{-c_2/2}P, e^{c_1/2}P],$$

where $P = (n_{i-1} \cdots n_{N-1})^{1/2}$.
Set $\phi = \min\{1 - e^{-c_1/2}, (e^{-c_2/2} - e^{-c_2})/(e^{-c_2/2} + e^{c_1})\}$. Take a $\phi$-net $\mathcal{N}_\phi$ for $S^{n_N - 1}$ with size $|\mathcal{N}_\phi| \leq (3/\phi)^{n_N}$. Notice that with this size we can actually cover the unit ball, not only the unit sphere.
Thus, with probability at least $1 - |\mathcal{N}_\phi| \exp\left\{-\Omega\left(\frac{1}{\Delta_1}\right)\right\}$, for all $u \in \mathcal{N}_\phi$ simultaneously we have

$$\|Au\|_2 / \|u\|_2 \in [e^{-c_2/2}P, e^{c_1/2}P].$$

Fixed $v \in S^{n_N - 1}$, there exists $u \in \mathcal{N}_\phi$ such that $\|u - v\|_2 \leq \phi$. WLOG, we assume $1 - \phi \leq \|u\|_2 \leq 1$. We obtain

$$\|Av\|_2 \leq \|Au\|_2 + \|A(u - v)\|_2 \leq e^{c_1/2}P + \phi\|A\| .$$

Taking supereme over $\|v\|_2 = 1$, we obtain

$$\sigma_{max}(A) = \|A\| \leq \frac{e^{c_1/2}}{1 - \phi}P \leq e^{c_1}P.$$

For the lower bound, we have

$$\|Av\|_2 \geq \|Au\|_2 - \|A(u - v)\|_2 \geq e^{-c_2/2}P\|u\| - \phi\|A\| \geq \left[(1 - \phi)e^{-c_2/2} - \phi e^{c_1}\right]P \geq e^{-c_2}P.$$

Taking the infimum over $\|v\|_2 = 1$, we get

$$\sigma_{min}(A) \geq e^{-c_2}P.$$

The conclusions hold with probability at least

$$1 - |\mathcal{N}_\phi| \exp\left\{-\Omega\left(\frac{1}{\Delta_1}\right)\right\}$$

$$\geq 1 - \exp\{n_N \ln(3/\phi)\} \exp\left\{-\Omega\left(\frac{1}{\Delta_1}\right)\right\}$$

$$\geq 1 - \exp\left\{-\Omega\left(\frac{1}{\Delta_1}\right)\right\},$$

since $n_N \Delta_1 \leq C(c_1, c_2)$. $\qquad\square$

**Lemma E.4.** *There exists a positive constant $C(c_1, c_2)$ which only depends on $c_1, c_2$, such that if $rank(X)\Delta_1 \leq C(c_1, c_2)$, then for any fixed $1 \leq j < N$, with probability at least $1 - \exp\{-\Omega\left(\frac{1}{\Delta_1}\right)\}$ we have*

$$\sigma_{\max}(W_{j:1}(0)|_{\mathcal{R}(X)}) \leq e^{c_1}(n_1 n_2 \cdots n_j)^{1/2}, \tag{62}$$

*and*

$$\sigma_{\min}(W_{j:1}(0)|_{\mathcal{R}(X)}) \geq e^{-c_2}(n_1 n_2 \cdots n_j)^{1/2}. \tag{63}$$

*Proof of Lemma E.4.* The proof is similar to that of previous lemma. The only difference is that now we consider the $\phi-$net to cover the unit sphere in $\mathcal{R}(X) \cap \mathbb{R}^{n_0}$, with $\dim \mathcal{R}(X) \cap \mathbb{R}^{n_0} = rank(X)$, where $\mathcal{R}(X)$ represents the column space of $X$. $\qquad\square$

**Lemma E.5.** *Set $C = n_{max}/n_{min} < \infty$, $\theta = 1/2$. Assume $\Omega(1/\Delta_1) \geq \frac{k}{\Delta_1}$, where $0 < k < 1$ is a constant and $\Delta_1$ satisfies*

$$\begin{cases} \Delta_1 \leq \min\left\{\frac{k}{5\ln(6)}, \frac{k}{5\ln(5\ln(6)e/k)}\right\} \\ \Delta_1 \ln(C) \leq \min\left\{\frac{k}{5\ln(5\ln(6)e/k)}, \frac{k}{5}\right\} \\ \Delta_1 \ln(N^{2\theta}) \leq k/5. \end{cases}$$

*Given $1 < i \leq j < N$, with probability at least $1 - 2e^{-k/(5\Delta_1)} = 1 - e^{-\Omega(1/\Delta_1)}$ we have*

$$\|W_{j:i}(0)\| \leq M_k\sqrt{C}N^\theta(n_i \cdots n_{j-1} \cdot \max\{n_{i-1}, n_j\})^{1/2},$$

*where $M_k$ is a positive constant that only depends on $k$.*

*Proof of Lemma E.5.* WLOG, assume $n_{i-1} \leq n_j$. Let $A = W_{j:i}(0)$. From lemma E.2, we know that fixed $v \in S^{n_{i-1}-1}$, with probability at least $1 - e^{-\Omega(1/\Delta_1)}$ we have $\|Av\|_2 \leq 4/3(n_i \cdots n_j)^{1/2}$.
.
Take a small constant $c = \frac{kN^{2\theta}}{5\ln(6)\Delta_1 n_{i-1}} \geq \frac{k}{5\ln(6)C}$. Let $v_1, \cdots, v_{n_{i-1}}$ be an orthonormal basis for $R^{n_{i-1}}$. Partition the index set $\{1, 2, \cdots, v_{n_{i-1}}\} = S_1 \cup S_2 \cup \cdots \cup S_{\lceil N^{2\theta}/c\rceil}$, where $|S_l| \leq \lceil cn_{i-1}/N^{2\theta}\rceil$ for each $1 \leq l \leq \lceil N^{2\theta}/c\rceil$.
The following discussion is similar to the proof of lemma E.3, hence we omit some details. For each $l$, taking a $1/2-$ net $\mathcal{N}_l$ for the set $V_{S_l} = \{v \in S^{n_{i-1}-1}; v \in span\{v_i; i \in S_l\}\}$, we can get

$$\|Au\|_2 \leq 4(n_i \cdots n_j)^{1/2}, u \in V_{S_l},$$

with probability at least

$$1 - |\mathcal{N}_l|e^{-k/\Delta_1} \geq 1 - \exp\{-k/\Delta_1 + (cn_{i-1}/N + 1)\ln 6\} \geq 1 - e^{-3k/(5\Delta_1)},$$

since $\Delta_1 \leq \frac{k}{5\ln(6)}$.
Therefore, for any $v \in \mathbb{R}^{n_{i-1}}$, we can write it as the sum $v = \sum_l a_l v_l$, where $\alpha_l \in \mathbb{R}$ and $v_l \in V_{S_l}$ for each $l$. We also know that $\|v\|_2^2 = \sum_{l \geq 1}|\alpha_l|^2$.
Then we have

$$\|Av\|_2 \leq \sum_l |\alpha_l|\,\|Av_l\|_2 \leq 4(n_i \cdots n_j)^{1/2}\sqrt{\lceil N^{2\theta}/c\rceil \sum_l |a_l|^2} \leq M_k\sqrt{C}N^\theta(n_i \cdots n_j)^{1/2}\,\|v\|_2.$$

Thus,

$$\|A\| \leq M_k\sqrt{C}N^\theta(n_i \cdots n_j)^{1/2}.$$

Notice that when $C \leq e$, $\Delta_1 \leq \frac{k}{5\ln(5\ln(6)e/k)} \leq \frac{k}{5\ln(5\ln(6)\cdot C/k)}$, and when $C > e$, we have

$$\Delta_1 \ln(C) \leq \min\left\{\frac{k}{5\ln(5\ln(6)e/k)}, k/5\right\} \leq \frac{k\ln(C)}{5\ln(5\ln(6)\cdot C/k)}.$$

The success probability is at least

$$1 - \lceil N^{2\theta}/c\rceil \cdot e^{-3k/(5\Delta_1)}$$
$$\geq 1 - \exp\left\{\ln\left(\frac{5\ln(6)\cdot C}{k}\right) + \ln(N^{2\theta}) - 3k/(5\Delta_1)\right\} - e^{-3k/(5\Delta_1)}$$
$$\geq 1 - 2e^{-k/(5\Delta_1)},$$

since

$$\Delta_1 \leq \frac{k}{5\ln\left(5\ln(6)\cdot C/k\right)} \text{ and } \Delta_1\ln(N^{2\theta}) \leq k/5.$$

$\square$

*Proof of Lemma 2.3.* Set $r = rank(X)$, and $u_1, \cdots, u_r$ be an orthonormal basis of the column space of $X$.

Then, $P_X = \sum_{i=1}^{r} u_i u_i^T$.

Notice that

$$\|a_n W_{N:1}(0)\|_X^2 = \|a_n W_{N:1}(0) P_X\|_F^2 = \sum_{i=1}^{r} \|a_n W_{N:1}(0) u_i\|_2^2.$$

By assumption, we have

$$\mathbb{E}\|a_n W_{N:1}(0)\|_X^2 = \mathbb{E}\sum_{i=1}^{r} \|a_n W_{N:1}(0) u_i\|_2^2 = r.$$

The Markov inequality implies

$$\mathbb{P}(\|a_n W_{N:1}(0)\|_X^2 \geq \frac{2r}{\delta}) \leq \frac{\delta}{2}.$$

Therefore, we can bound the initial loss value as

$$
\begin{aligned}
L_0 - L(W_*) &\leq \langle \nabla L(W_*), a_N W_{N:1}(0) X - W_* \rangle + \frac{\beta}{2} \|a_N W_{N:1}(0) - W_*\|_X^2 \\
&= \frac{\beta}{2} \|a_N W_{N:1}(0) - W_*\|_X^2 \\
&\leq \beta(\|a_N W_{N:1}(0)\|_X^2 + \|W_*\|_X^2) \\
&\leq \beta(\frac{2r}{\delta} + \|W_*\|_X^2),
\end{aligned}
$$

with probability at least $1 - \delta/2$.

$\square$

*Proof of Theorem B.1.* The requirement on size $\{n_1, n_2, \cdots, n_{N-1}, N\}$ in (16) ensures that lemma E.3, E.4, E.5, 2.3, and D.1 hold.
WLOG, we set $c_1 = c/6, c_2 = c/3, M = 2M_k \sqrt{C_0}, B_0 = B_\delta$, and $\eta =: \frac{(1-\varepsilon)2n_N}{e^{2c}\beta N}$, then with probability at least

$$1 - N^2 e^{-\Omega(1/\Delta_1)} - \delta/2 \geq 1 - \delta, \text{ since } \Delta_1 \leq \frac{1}{C(c)} \min\left\{\frac{1}{\ln N}, \frac{1}{\ln(1/\delta)}\right\},$$

the random initialization satisfies the initialization assumption (31) and the overparameterization assumption (32). By applying Lemma D.1, we complete the proof.

$\square$

## F  ORTHOGONAL INITIALIZATION FALL INTO THE CONVERGENCE REGION

The following are some basic facts for random projection and embedding. Most of the following properties can be found in Eaton (1989).

**Proposition F.1.**

1. *A is a random embedding if and only if $A^T$ is a random projection.*

2. *If A is a square matrix, then random projection, random embedding and random orthogonal matrix are equivalent.*

3. *The uniform distribution on the group is a left and right invariant probability measure: that is, if A is a random orthogonal matrix, then $A, UA, AU$ are all random orthogonal matrices, where U is a non-random orthogonal matrix.*

4. *Assume $X$ is a $n \times q(q \leq n)$ random matrix whose entries are i.i.d. $N(0,1)$ random variables. Then $A := X(X^T X)^{-1/2}$ is a random embedding, since $A^T A = I_q$ and the distribution of $A$ is left invariant, which means that $A$ and $UA$ have the same distribution, where $U$ is a non-random orthogonal matrix.*

5. *If $A$ is a uniform distribution over an orthogonal group of order $n$ and $A$ is partitioned as $A = (A_1, A_2)$, where $A_1$ is $n \times q$ and $A_2$ is $n \times (n-q)$, then $A_1^T$ and $A_2^T$ are both random orthogonal matrix.*

6. *The columns of uniform distribution over the orthogonal group of order $n$, and*

$$\frac{(\xi_1, \cdots, \xi_n)}{\sqrt{\xi_1^2 + \xi_2^2 + \cdots + \xi_n^2}}$$

   *have the same distribution, where $\xi_1, \cdots, \xi_n$ are i.i.d. $N(0,1)$ random variables.*

7. *Assume $A = A_{n \times p}, n \leq p$ is a random orthogonal projection. For any $v \in S^{p-1}$, $\|Av\|_2^2$ and $(\sum_{i=1}^{n} \xi_i^2)/(\sum_{j=1}^{p} \xi_j^2)$ are both following beta distribution with $\alpha = n/2, \beta = (p - n)/2$, where $\xi_1, \cdots, \xi_n$ are i.i.d. $N(0,1)$ random variables.*

*Remark* 9. There are several ways to construct the random matrix $A = (a_{ij})_{q \times n}$, $q \leq n$, which is uniformly distributed over rectangular matrices with $AA^T = c^2 I_q, c > 0$. Let $O_n$ be uniformly distributed over a real orthogonal group of order n, and $O_n$ is partitioned as $O_n = (A_1^T, A_2^T)^T$, where $A_1$ is $q \times n$. Assume $X = (x_{ij})_{q \times n}$, and $x_{ij}$ are independent standard normal random variables. Then, $A$, $cA_1$, and $c(XX^T)^{-1/2}X$ have the same distribution.

**Lemma F.2.** *For any $x \in \mathbb{R}^{n_0}$, the one peak random projection and embedding initiation satisfies*

$$\mathbb{E}\left[\|a_N W_{N:1}(0)x\|_2^2\right] = \|x\|_2^2.$$

*Proof.* Let $D = W_{p:1}(0)/\sqrt{n_1 n_2 \cdots n_p}$. Then $D$ is an embedding matrix. Thus, $\|Dx\|_2^2 = \|x\|_2^2$. Let $A_i = W_{i:p+1}(0)/\sqrt{n_p n_{p+1} \cdots n_{i-1}}$, where $i \geq p+1$, and $A_p = I$.

Set $B_i = \|A_i Dx\|_2^2 / \|A_{i-1} Dx\|_2^2, i \geq p+1$. Then, $B_i$ follows beta distribution $B(n_i/2, (n_{i-1} - n_i)/2)$ given $B_{i-1}, B_{i-2}, \cdots, B_{p+1}, i \geq p+1$. If $n_i = n_{i-1}$, then $B_i|(B_{i-1}, B_{i-2}, \cdots, B_{p+1}) = 1$, a.s.

If $B \sim B(a,b)$, then the expectation is given by the following equation,

$$\mathbb{E}B = \frac{a}{a+b}.$$

Thus, by the law of total expectation, we have

$$\frac{n_N}{n_p}\mathbb{E}\|a_N W_{N:1}(0)x\|_2^2 = \mathbb{E}\|A_N Dx\|_2^2 = \mathbb{E}B_N B_{N-1} \cdots B_{p+1} \|Dx\|_2^2 = \frac{n_N}{n_p}\|x\|_2^2.$$

This completes the proof. $\qquad\square$

Next, we introduce sub-Gaussian random variables, associated with bounds on how a random variables deviate their expected value.

**Definition F.1.** A random variable $X$ with finite mean $\mu = \mathbb{E}X$ is sub-Gaussian if there is a positive number $\sigma$ such that:

$$\mathbb{E}[\exp(\lambda(X - \mu))] \leq \exp\left(\frac{\lambda^2\sigma^2}{2}\right) \text{ for all } \lambda \in \mathbb{R} \tag{64}$$

Such a constant $\sigma^2$ is called a proxy variance, and we say that $X$ is $\sigma^2$-sub-Gaussian, and we write $X \sim SG(\sigma^2)$.

**Example F.1.** Normal distribution $N(\mu, \sigma^2)$ of course is $\sigma^2$ sub-Gaussian.
For beta distribution, Elder (2016) showed that $B(a,b)$ is $\frac{1}{4(a+b)+2}$-sub-Gaussian and later, Marchal & Arbel (2017) concluded $\frac{1}{4(a+b+1)}$-sub-Gaussian.

The Hoeffding bound for random variable $X$ with a mean $\mu$ and sub-Gaussian parameter $\sigma$ is given by,

$$\mathbb{P}\left[|X - \mu| \geq t\right] \leq 2 \exp\left\{-\frac{t^2}{2\sigma^2}\right\}, \forall t \geq 0. \tag{65}$$

Simply applying the Chernoff bound for $B(a, b)$, we obtain the following lemma.

**Lemma F.3.** *Assume a random variable $B$ distributed as a beta distribution $B(a, b)$ with two positive shape parameters $a$ and $b$. Then,*

$$\mathbb{P}\left(\left|B - \frac{a}{a+b}\right| \geq y\right) \leq 2 \exp\left\{-2(a+b)y^2\right\}, y \geq 0.$$

*Hence,*

$$\mathbb{P}\left(\left|B - \frac{a}{a+b}\right| \leq \varepsilon \frac{a}{a+b}\right) \geq 1 - \exp\{-\Omega(a^2/(a+b))\},$$

*where $\Omega(\cdot)$ only depend on $\varepsilon$.*
*For the upper tail, we can obtain a better bound,*

$$\mathbb{P}\left(B \geq (1 + \varepsilon)\frac{a}{a+b}\right) \leq \exp\left\{-(\varepsilon - \ln(\varepsilon + 1))a\right\}. \tag{66}$$

*Proof of Lemma F.3.* We only need to prove the third inequality. Assume random variable $B \sim B(a, b)$. Set $v = a + b$, $(1 + t)\frac{a}{v} \leq y < 1, t > 0$, and $r > 0$.
We are going to estimate the Chernoff bound for $B$, which is

$$\mathbb{P}(B \geq y) \leq e^{-(ry - \ln \mathbb{E}e^{rB})} =: e^{-I_r(y)}.$$

The moment generating function of $B$ is given by

$$\mathbb{E}e^{rB} = 1 + \sum_{k=1}^{\infty} \frac{a(a+1)\cdots(a+k-1)}{v(v+1)\cdots(v+k-1)} \frac{r^k}{k!} \leq 1 + \sum_{k=1}^{\infty} \frac{a(a+1)\cdots(a+k-1)}{v^k} \frac{r^k}{k!}, r > 0.$$

Recall that the Maclaurin series of $(1 - r/v)^{-a}$ over $(-v, v)$, is given by equation

$$(1 - r/v)^{-a} = 1 + \sum_{k=1}^{\infty} \frac{a(a+1)\cdots(a+k-1)}{v^k} \frac{r^k}{k!}.$$

Thus,

$$I_r(y) = ry - \ln \mathbb{E}e^{rB} \geq ry + a\ln(1 - r/v).$$

Set $r = v - a/y \in (0, v)$. We obtain

$$\mathbb{P}(B \geq y) \leq \exp\{-(vy - a + a\ln(a/(vy)))\} =: \exp\{-vy \cdot g(a/(vy))\}, (1 + t)\frac{a}{v} \leq y < 1$$

where $g(x) = 1 - x + x\ln(x)$, $x = a/(vy) \in (0, 1/(1 + t)]$. Notice that $g(1) = 0$ and $g'(x) = \ln(x) < 0$ over $x \in (0, 1)$.
We know that

$$g(x) \geq g(1/(1 + t)) = \frac{t - \ln(1 + t)}{t + 1}, t > 0.$$

Thus,

$$\mathbb{P}(B \geq y) \leq \exp\left\{-vy \cdot \frac{t - \ln(1 + t)}{t + 1}\right\} = \exp\left\{-(t - \ln(1 + t))a\right\}, y = (1 + t)\frac{a}{v} < 1.$$

Set $y = (1 + \varepsilon)\frac{a}{a+b}$. We obtain the inequality (66). $\square$

*Remark* 10. It is trivial to check
$\|W_{j:i}(0)\| = (n_i n_{i+1} \cdots n_j)^{1/2}, 1 \leq i \leq j \leq p,$
$\|W_{j:i}(0)\| = (n_{i-1} n_i \cdots n_{j-1})^{1/2}, p + 1 \leq i \leq j \leq N,$
$\|W_{j:i}(0)\| \leq (n_i n_{i+1} \cdots n_{j-1})^{1/2} (n_p)^{1/2}$

$$\leq \left(\frac{n_{max}}{n_{min}}\right)^{1/2} (n_i n_{i+1} \cdots n_{j-1} \cdot \max\{n_{i-1}, n_j\})^{1/2}, 1 \leq i < p < j \leq N, (i, j) \neq (1, N).$$

*Remark* 11. As a special case, if $n_1 = n_2 = \cdots = n_{N-1} = n$, we know that $\|W_{j:i}(0)\| = (n_{i-1}n_i \cdots n_{N-1})^{1/2} = n^{(N-i+1)/2}$.

**Lemma F.4.** *Assume $n_p/\min\{n_1, n_{N-1}\} \leq C_0 < \infty$. Set $\varepsilon > 0$. Let $C(\varepsilon)$ represent the constant depend only on $\varepsilon$. If $n_1/C_0 \geq C(\varepsilon)n_N$, then with probability at least $1 - e^{-\Omega(n_1/C_0)}$*

$$\sigma_{max}(W_{N:i}(0)) \leq (1+\varepsilon)(n_{i-1}n_i \cdots n_{N-1})^{1/2}, 2 \leq i \leq p$$

$$\sigma_{min}(W_{N:i}(0)) \geq (1-\varepsilon)(n_{i-1}n_i \cdots n_{N-1})^{1/2}, 2 \leq i \leq p.$$

*Similarly, if $n_{N-1}/C_0 \geq C(\varepsilon)rank(X)$, then with probability at least $1 - e^{-\Omega(n_{N-1}/C_0)}$*

$$\sigma_{max}(W_{j:1}(0)|_{\mathcal{R}(X)}) \leq (1+\varepsilon)(n_1 n_2 \cdots n_j)^{1/2}, p+1 \leq j \leq N$$

$$\sigma_{min}(W_{j:1}(0)|_{\mathcal{R}(X)}) \geq (1-\varepsilon)(n_1 n_2 \cdots n_j)^{1/2}, p+1 \leq j \leq N.$$

*Proof of Lemma F.4.* Let $D = (n_{N-1}n_{N-2} \cdots n_p)^{-1/2}W_{N:p+1}^T(0)$ and $A_i = (n_p n_{p-1} \cdots n_i)^{-1/2}W_{p:i}^T(0)$. Assume $v \in S^{n_N-1}$. Easy to see that $A_i$ is a product of random orthogonal projection and $D$ is a random embedding.
Let $e_1 = (1, 0, 0, \cdots, 0)^T \in \mathbb{R}^{n_p}$. There exists orthogonal matrix $T$ such that $TDv = e_1$, $\|e_1\|_2 = \|TDv\|_2 = \|v\|_2 = 1$.
Since random orthogonal projection are right invariant, we have

$$\mathbb{P}(\|A_iDv\|_2 \geq y) = \mathbb{E}\left[\mathbb{E}\left(I_{\{\|A_iT^Te_1\|_2 \geq y\}}\Big| D\right)\right] = \mathbb{E}\left[\mathbb{E}\left(I_{\{\|A_ie_1\|_2 \geq y\}}\Big| D\right)\right] = \mathbb{P}(\|A_ie_1\|_2 \geq y).$$

This proves that $\|A_iDv\|_2^2$ and $\|A_ie_1\|_2^2$ have the same distribution.

**Claim**: If $v \neq 0$, then $\|A_iDv\|_2^2 / \|v\|_2^2 = \left\|(n_i n_{i+1} \cdots n_p^2 \cdots n_{N-1})^{-1/2}W_{N:i}^T v\right\|_2^2 / \|v\|_2^2$ follows beta distribution $B(n_{i-1}/2, (n_p - n_{i-1})/2)$.
Define $B_p = \|A_pe_1\|_2^2$, $B_i = \|A_ie_1\|_2^2 / \|A_{i+1}e_1\|_2^2$, $i = p-1, p-2, \cdots, 1$.
Then $B_p \sim B(n_{p-1}/2, (n_p - n_{p-1})/2)$, $B_{p-1}|B_p \sim B(n_{p-2}/2, (n_{p-1} - n_{p-2})/2)$, $\cdots$, $B_i|(B_p, \cdots, B_{i+1}) \sim B(n_{i-1}/2, (n_i - n_{i-1})/2)$.
If $n_{i+1} = n_i$, we know that $B_i|(B_p, \cdots, B_{i+1}) = 1, a.s.$
If $B \sim B(a, b)$, then the moments are given by the following equations,

$$\mathbb{E}B = \frac{a}{a+b}, \text{ and } \mathbb{E}B^k = \frac{a}{a+b}\frac{a+1}{a+b+1} \cdots \frac{a+k-1}{a+b+k-1}. \tag{67}$$

By the law of total expectation, we have

$$\mathbb{E}B_iB_{i+1} \cdots B_p = \frac{n_{i-1}}{n_i}\frac{n_i}{n_{i+1}} \cdots \frac{n_{p-1}}{n_p} = \frac{n_{i-1}}{n_p},$$

as well as

$$\mathbb{E}(B_iB_{i+1} \cdots B_p)^k = \frac{n_{i-1}/2}{n_p/2}\frac{n_{i-1}/2+1}{n_p/2+1} \cdots \frac{n_{i-1}/2+k-1}{n_p/2+k-1}.$$

Notice that all the integer moments of $B_iB_{i+1} \cdots B_p$ match those of $B(n_{i-1}/2, (n_p - n_{i-1})/2)$. We can verify that beta distribution satisfies Carleman's condition, which implies that $B_iB_{i+1} \cdots B_p \sim B(n_{i-1}/2, (n_p - n_{i-1})/2)$.
Thus, $\|A_iDv\|_2^2 / \|v\|_2^2 \sim B(n_{i-1}/2, (n_p - n_{i-1})/2)$, which proves the claim.

With probability at least $1 - \exp\{-\Omega(n_1/C_0)\}$, we have

$$(1-\varepsilon)^2\frac{n_{i-1}}{n_p} \leq \|ADv\|_2^2 \leq (1+\varepsilon)^2\frac{n_{i-1}}{n_p}, \|v\|_2 = 1.$$

Using the $\phi-$net technique, which has been already used to prove lemma E.3, we know that

$$\sigma_{min}(AD) \geq (1-\varepsilon)\left(\frac{n_{i-1}}{n_p}\right)^{1/2},$$

and

$$\sigma_{max}(AD) \leq (1+\varepsilon)\left(\frac{n_{i-1}}{n_p}\right)^{1/2},$$

with probability at least $1 - \exp\{n_N \ln(3/\phi(\varepsilon))\} \exp\{-\Omega(n_1/C_0)\} \geq 1 - \exp\{-\Omega(n_1/C_0)\}$, since $n_1/C_0 \geq C(\varepsilon)n_N$, for $2 \leq i \leq p$.

Hence, with probability at least $1 - e^{-\Omega(n_1/C_0)}$, we have

$$\sigma_{min}(W_{N:i}(0)) \geq (1 - \varepsilon)(n_{i-1} \cdots n_{N-1})^{1/2},$$

and

$$\sigma_{max}(W_{N:i}(0)) \leq (1 + \varepsilon)(n_{i-1} \cdots n_{N-1})^{1/2}.$$

The other part of the proof is similar to that of lemma E.4, so we omit it.

$\square$

*Proof of Theorem B.2*. Set $c > 0$, $c_1 = c/6, c_2 = c/3$. In lemma F.4, we can pick a $\varepsilon > 0$, such that $1 + \varepsilon \leq e^{c_1/2}$ and $1 - \varepsilon \geq e^{-c_2/2}$. Set $M = 2\sqrt{C_0}, \theta = 0, B_0 = B_\delta$, and $\eta = \frac{(1-\varepsilon)2n_N}{e^{2c}\beta N}$.

The requirement on size $\{n_1, n_2, \cdots, n_{N-1}, N\}$ in (17) make sure that the remark 10, lemma F.4, lemma 2.3, and lemma D.1 all hold.

Notice that even though we need the conclusions in lemma F.4 to hold simultaneously for $2 \leq i \leq p$, $p + 1 \leq j \leq N$, it suffices to apply lemma F.4 over $i \in I$ and $j \in J$, such that $\{n_i; i \in I\}$ and $\{n_j; j \in I\}$ both have distinct values. Since $|I| \leq \underline{N}$ and $|J| \leq \underline{N}$, with probability at least

$$1 - 2\underline{N}e^{-\Omega(n_{min}/C_0)} - \delta/2 \geq 1 - \delta,$$

the one peak random orthogonal projection and embedding initialization satisfies the initialization assumption (31) and the overparameterization assumption (32).

Under the assumption $n_1 = n_2 = \cdots = n_{N-1}$, we can use remark 11 to replace lemma F.4. Thus, with probability at least $1 - \delta/2 \geq 1 - \delta$, (31) holds. By applying lemma 2.3 and D.1, we complete the proof.

$\square$

*Proof of Theorem B.3*. Let $W_N(0) = \sqrt{n}U_N[I_{n_y}, 0]V_N^T, \cdots, W_i(0) = \sqrt{n}U_i I_n V_i^T, 2 \leq i \leq N - 1$, and $W_1(0) = \sqrt{n}U_1[I_{n_x}, 0]^T V_1^T$. Now, we want to verify (31). By a simple calculation, we have

$$\begin{cases} \sigma_{max}(W_{N:i+1}(0)) = \sigma_{min}(W_{N:i+1}(0)) = n^{(N-i)/2}, 1 \leq i \leq N - 1, \\ \sigma_{max}(W_{i-1:1}(0)|_{\mathcal{R}(X)}) = \sigma_{max}(W_{i-1:1}(0)|_{\mathcal{R}(X)}) = n^{(i-1)/2}, 2 \leq i \leq N, \\ \|W_{j:i}(0)\| = n^{(j-i+1)/2}, 1 < i \leq j < N. \end{cases} \quad (68)$$

Notice that for any $1 \leq p \leq m$

$$\|a_N W_{N:1}(0)x\|_2^2 = \frac{n}{n_N}\left\|U_N[I_{n_y}, 0]V_N^T U_N[I_{n_x}, 0]^T V_1^T x\right\|_2^2 = \frac{n}{n_N}\left\|U_N[I_{n_y}, 0]V_N^T x'\right\|_2^2,$$

where $x' = U_N[I_{n_x}, 0]^T V_1^T x, \|x\|_2 = \|x'\|_2$.

Since the distribution of $U_N[I_{n_y}, 0]V_N^T$ is right invariant under multiplying orthogonal matrices, we have

$$\left\|U_N[I_{n_y}, 0]V_N^T x'\right\|_2^2 / \|x\|_2^2 \sim B(\frac{n_y}{2}, \frac{n - n_y}{2}).$$

Thus,

$$\mathbb{E}\left[\|a_N W_{N:1}(0)x\|_2^2\right] = \|x\|_2^2.$$

Applying lemma 2.3, we have

$$L_0 - L(W_*) \leq \beta\left(\frac{2 \cdot rank(X)}{\delta} + \|W_*\|_X^2\right),$$

with probability at least $1 - \delta/2$.

By applying Lemma D.1 with $c > 0, c_1 = c/6, c_2 = c/3, \theta = 0$, we complete the proof. $\square$

*Proof of Theorem 3.1*. Theorem 3.1 is a special case of Theorem B.1 and Theorem B.2. Hence, we omit the proof. $\square$

*Proof of Theorem 3.2.* In Theorem B.1, B.2, and B.3, we proved that for given constant $c_1, c_2 > 0$ and $0 < \varepsilon, \delta/2 < 1/2$ as well as a learning rate $\eta$, there exists a constant $C = C(c_1, c_2)$ such that all three kinds of random initializations will fall into the convergence region with probability at least $1 - \delta$. By applying Lemma 2.3, we complete the proof.

$\square$

## G  TABLES

In this section, we provide some empirical evidences to support the argument expressed in Section 4:**Why do bad saddles not affect GD for overparameterized deep linear neural networks?** Consider the following procedures for tables of $\frac{\|W_i(t) - W_i(0)\|_F}{\|W_i(0)\|_F}$:

a) We consider $X \in \mathbb{R}^{128 \times 1000}$, and $W_* \in \mathbb{R}^{10 \times 128}$ and set $Y = W_* X + \varepsilon$, where the entries in $X$ and $\varepsilon$ are drawn i.i.d. from $N(0, 1)$.

b) We consider the loss function $\frac{1}{2} \|a_N W_{N:1} X - Y\|_F^2$.

c) For the given deep linear networks, we apply the orthogonal initializations, which are denoted as $W_j(0), 1 \le j \le N$.

d) We set the learning rate as $\eta = \frac{n_N}{N \cdot \|X\|^2}$ for the deep linear neural networks.

e) We prepare the tables for $\frac{\|W_i(t) - W_i(0)\|_F}{\|W_i(0)\|_F}$.

Let $n_1 = n_2 = n_3 = 2000, N = 4$. Assume $W_*$ are drawn i.i.d. from $N(0, 25)$. We obtain the following table:

|          | $i = 1$ | $i = 2$ | $i = 3$ | $i = 4$ |
|----------|---------|---------|---------|---------|
| $t = 1$  | 0.05161 | 0.00826 | 0.00826 | 0.18464 |
| $t = 2$  | 0.08779 | 0.01389 | 0.01389 | 0.31396 |
| $t = 3$  | 0.11335 | 0.01781 | 0.01779 | 0.40435 |
| $t = 4$  | 0.12109 | 0.01894 | 0.01889 | 0.42920 |
| $t = 5$  | 0.12527 | 0.01956 | 0.01948 | 0.44282 |
| $t = 6$  | 0.12611 | 0.01967 | 0.01958 | 0.44476 |
| $t = 7$  | 0.12755 | 0.01988 | 0.01978 | 0.44955 |
| $t = 8$  | 0.12745 | 0.01986 | 0.01975 | 0.44876 |
| $t = 9$  | 0.12819 | 0.01997 | 0.01987 | 0.45136 |
| $t = 10$ | 0.12793 | 0.01992 | 0.01982 | 0.45018 |

Let $n_1 = n_2 = 10000, N = 2$. Assume $W_*$ are drawn i.i.d. from $N(0, 4)$. We obtain the following table:

|          | $i = 1$ | $i = 2$ | $i = 3$ |
|----------|---------|---------|---------|
| $t = 1$  | 0.02708 | 0.00153 | 0.04844 |
| $t = 2$  | 0.04319 | 0.00244 | 0.07727 |
| $t = 3$  | 0.05296 | 0.00299 | 0.09474 |
| $t = 4$  | 0.05888 | 0.00333 | 0.10533 |
| $t = 5$  | 0.06248 | 0.00353 | 0.11176 |
| $t = 6$  | 0.06468 | 0.00365 | 0.11569 |
| $t = 7$  | 0.06603 | 0.00373 | 0.11811 |
| $t = 8$  | 0.06688 | 0.00377 | 0.11962 |
| $t = 9$  | 0.06741 | 0.00380 | 0.12057 |
| $t = 10$ | 0.06775 | 0.00382 | 0.12117 |

Let $n_1 = n_2 = 4000, N = 2$. Assume $W_*$ are drawn i.i.d. from $N(0, 1)$. We obtain the following table:

|         | $i = 1$ | $i = 2$ | $i = 3$ |
|---------|---------|---------|---------|
| $t = 1$  | 0.01622 | 0.00290 | 0.05802 |
| $t = 2$  | 0.02684 | 0.00480 | 0.09601 |
| $t = 3$  | 0.03411 | 0.00609 | 0.12202 |
| $t = 4$  | 0.03919 | 0.00700 | 0.14018 |
| $t = 5$  | 0.04280 | 0.00764 | 0.15306 |
| $t = 6$  | 0.04539 | 0.00810 | 0.16232 |
| $t = 7$  | 0.04729 | 0.00844 | 0.16908 |
| $t = 8$  | 0.04869 | 0.00868 | 0.17408 |
| $t = 9$  | 0.04974 | 0.00887 | 0.17782 |
| $t = 10$ | 0.05054 | 0.00901 | 0.18066 |

Let $n_1 = n_2 = 8000, N = 2$. Assume $W_*$ are drawn i.i.d. from $N(0, 1)$. We obtain the following table:

|         | $i = 1$ | $i = 2$ | $i = 3$ |
|---------|---------|---------|---------|
| $t = 1$  | 0.01173 | 0.00148 | 0.04195 |
| $t = 2$  | 0.01944 | 0.00246 | 0.06955 |
| $t = 3$  | 0.02470 | 0.00312 | 0.08838 |
| $t = 4$  | 0.02838 | 0.00358 | 0.10151 |
| $t = 5$  | 0.03098 | 0.00391 | 0.11083 |
| $t = 6$  | 0.03287 | 0.00415 | 0.11758 |
| $t = 7$  | 0.03426 | 0.00432 | 0.12253 |
| $t = 8$  | 0.03530 | 0.00445 | 0.12624 |
| $t = 9$  | 0.03608 | 0.00455 | 0.12904 |
| $t = 10$ | 0.03668 | 0.00463 | 0.13118 |

Let $n_1 = n_2 = 12000, N = 2$. Assume $W_*$ are drawn i.i.d. from $N(0, 1)$. We obtain the following table:

|         | $i = 1$ | $i = 2$ | $i = 3$ |
|---------|---------|---------|---------|
| $t = 1$  | 0.00965 | 0.00099 | 0.03453 |
| $t = 2$  | 0.01597 | 0.00164 | 0.05712 |
| $t = 3$  | 0.02025 | 0.00208 | 0.07244 |
| $t = 4$  | 0.02323 | 0.00239 | 0.08310 |
| $t = 5$  | 0.02535 | 0.00261 | 0.09069 |
| $t = 6$  | 0.02690 | 0.00277 | 0.09621 |
| $t = 7$  | 0.02804 | 0.00289 | 0.10029 |
| $t = 8$  | 0.02890 | 0.00297 | 0.10336 |
| $t = 9$  | 0.02955 | 0.00304 | 0.10570 |
| $t = 10$ | 0.03006 | 0.00309 | 0.10750 |

Let $n_1 = n_2 = 20000, N = 2$. Assume $W_*$ are drawn i.i.d. from $N(0, 1)$. We obtain the following table:

|         | $i = 1$ | $i = 2$ | $i = 3$ |
|---------|---------|---------|---------|
| $t = 1$  | 0.00713 | 0.00057 | 0.02551 |
| $t = 2$  | 0.01181 | 0.00095 | 0.04225 |
| $t = 3$  | 0.01499 | 0.00121 | 0.05362 |
| $t = 4$  | 0.01720 | 0.00138 | 0.06154 |
| $t = 5$  | 0.01878 | 0.00151 | 0.06720 |
| $t = 6$  | 0.01994 | 0.00161 | 0.07132 |
| $t = 7$  | 0.02079 | 0.00168 | 0.07438 |
| $t = 8$  | 0.02144 | 0.00173 | 0.07668 |
| $t = 9$  | 0.02193 | 0.00177 | 0.07844 |
| $t = 10$ | 0.02231 | 0.00179 | 0.07981 |

## H  FIGURES

In this section, we provide some empirical evidences to support the results in Section 4: **Numerical Experiments**. We will show how the trajectories of the non-convex deep linear neural networks are

related to a convex optimization problem for GD under different initialization schemes. Consider the following procedures for plots of the logarithm of loss as a function of number of iterations:

a) We choose $X \in \mathbb{R}^{128 \times 1000}$ and $W_* \in \mathbb{R}^{10 \times 128}$ and set $Y = W_* X + \varepsilon$, where the entries in $X$, $W_*$ and $\varepsilon$ are drawn i.i.d. from $N(0,1)$.

b) We consider the loss function $\frac{1}{2} \|a_N W_{N:1} X - Y\|_F^2$.

c) For the given linear networks, we apply the Gaussian initialization and one peak random orthogonal projection and embedding initialization, which are denoted as $W_j(0), 1 \leq j \leq N$.

d) For the convex optimization problem (1), we set the initialization to be $W(0) = a_N W_N(0) \cdots W_1(0)$.

e) We set the learning rates as $\eta = \frac{n_N}{N \cdot \|X\|^2}$ and $\eta_* = \frac{N}{n_N} \eta$ for the deep linear neural networks and convex problem, respectively.

f) We draw the loss function through 25 iterations.

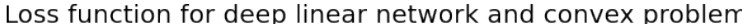

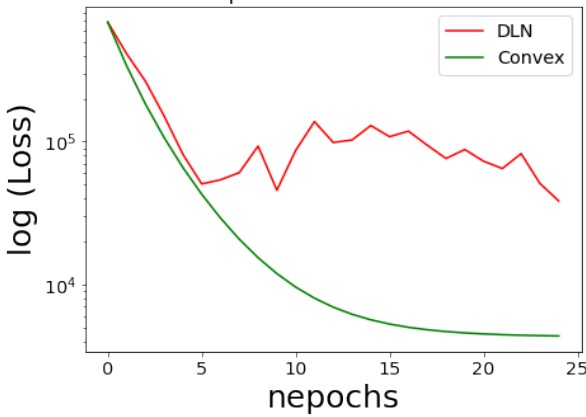

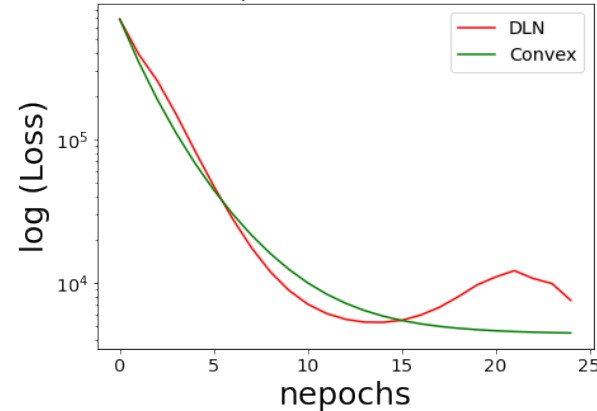

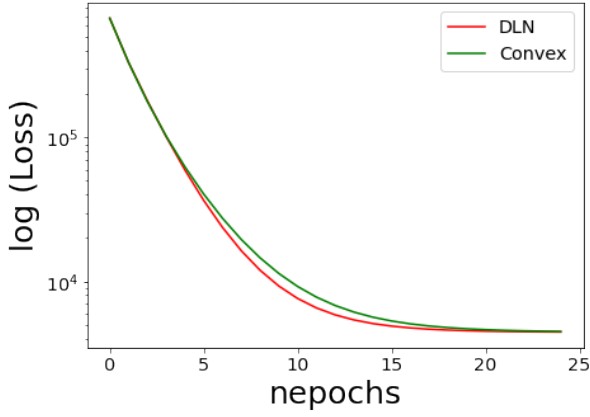

Figure 1: Plot of Loss as a function of number of iterations with $n_1 = n_2 = n_3 = 128$ (First), 200 (Second), 2000 (Third) for Gaussian initialization, respectively.

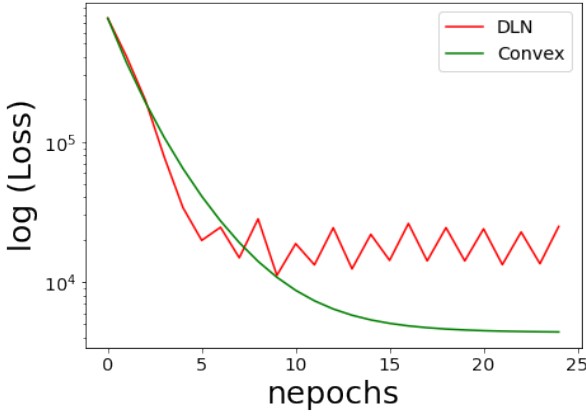

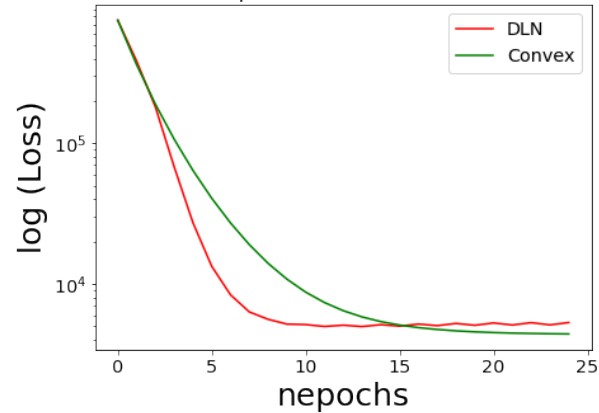

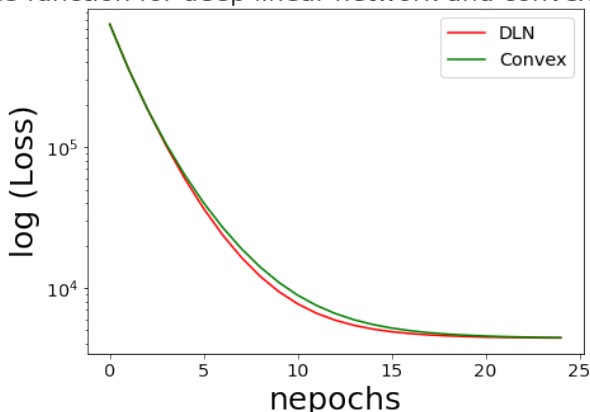

Figure 2: Plot of Loss as a function of number of iterations with $n_1 = n_2 = n_3 = 128$ (First), 200 (Second), 5000 (Third) for Orthogonal initialization, respectively.

