# OpenReview forum: "Sharp Convergence Analysis of Gradient Descent for Deep Linear Neural Networks"
_ICLR.cc/2023/Conference — Submitted to ICLR 2023_

### Official Review · Reviewer_gRRu · 2022-10-20

**Confidence:** 3
**Clarity, Quality, Novelty And Reproducibility:** Both clarity and novelty are good
**Correctness:** 3
**Technical Novelty And Significance:** 3
**Empirical Novelty And Significance:** Not applicable
**Recommendation:** 6

**Strength And Weaknesses:**

Strength: (1) the result has some fundamental importance. (2) the result that deeper nets can converge as fast as a convex linear model is both rather surprising and important

I think the main weakness is the lack of insights and discussion. The following problems should be addressed by the authors before publication:

(1) a few overclaims: the analysis only revolves around overparametrized models, thus the title should also include the word "overparametrization"
- for example, the paper did not obtain any result when the width of the model is small, whereas the title seems to suggest that the results apply to both cases
- the same criticism applies to the abstract
- the authors need to update both the title and the abstract to be more specific and avoid overclaiming

(2) the minimum width requirement in theorem 3.1 and 3.2 depends on constants $C, C_2, C_5$. These do not affect the convergence rate, but I think it will provide more insight if the authors provide a remark discussing how they depend on the hyperparameters of the problem, such as the depth of the model, the distribution of the data, etc

(3) a crucial distinction that the authors make is that overparametrization is essential for achieving such a good convergence rate; however, this point is way under-discussed. The authors should spend a lot more energy discussing why overparametrization is essential
- for example, the authors could explain why the results/theoretical techniques cannot be applied to thinner models and then point out or suggest what properties of a thin model prevent it from being efficiently optimized

(4) in the related works, the authors mentioned the previous results that deeper models can have vanishing Hessian that hinders optimization. The authors should then explain why this is not a problem from their theory
- for example, what properties of an overparametrized model (or what assumptions that the authors made) help escape such saddles

(5) how does the result change if there is a regularization term such as $L_2$? the authors should also discuss this relevant work: https://arxiv.org/abs/2202.04777

(6) does the main result change in a minibatch setting? the authors should suggest some answer

(7) the font size in the figures is too small. The  authors need to update it to match the font size of the main text

**Summary Of The Paper:**

This work proves that overparametrized deep linear networks can converge as fast as gradient descent on the equivalent linear regressor

**Summary Of The Review:**

I recommend rejecting the current version due to the lack of insight and discussions

However, I think the problems can be easily fixed and I would like to change my assessment if the authors make the corresponding improvements

---

> ### Author Response · Authors · 2022-11-19
> **Thanks for your careful review and insightful question**
>
> Thank you for valuable comments, and questions. Please see below for our responses to each individual question.
>
> Q1: a few overclaims:
>
> Thanks for your reasonable suggestion. We changed our title to Sharp Convergence Analysis of Gradient Descent for Overparameterized Deep Linear Neural Networks, and changed our abstract as well to avoid overclaims.
>
> Q2: the minimum width requirement in theorem 3.1 and 3.2 depends on constants $C, C_1, C_5 $ $\cdots$
>
> We add remark 6 to our latest version of the paper which answers how these constants depend on hyperparameters.
>
> Q3: overparametrization is essential for achieving such a good convergence rate; however, this point is way under-discussed:
>
> Thanks for your critical observations. We add a new section 4 in our latest version of the paper to describe the insights behind our main results. In this new section, we answer the following questions: Why GD trajectories for overparameterized deep linear neural networks with approximate balanced initialization are close to those for convex problems? Why do bad saddles not affect GD for overparameterized deep linear neural networks? Why does the width have to be large?
>
> Q4: what properties of an overparametrized model (or what assumptions that the authors made) help escape such saddles?
>
> You have a very good point. A complete answer for this question can be found in section 4: Why do bad saddles not affect GD for overparameterized deep linear neural networks? A short answer is that the overparametrization around balanced initialization avoids bad saddles through the whole training process. The main results in Kawaguchi (2016) imply that for $L_2$ loss, the bad saddles always satisfy that $W_{N-1:2}$ is not full rank; and through the GD training, we can show that $\inf_{t}\sigma_{min}(W_{N-1:2}(t))>0$. A deeper reasoning is related to section 4: Why does the width have to be large?
>
> Q5: how does the result change if there is a regularization term such as $L_2$? the authors should also discuss this relevant work: https://arxiv.org/abs/2202.04777
>
> Thanks for pointing out this relevant work, which might shed light on the GD convergence analysis for deep linear networks (probably with $L_2$ regularization) with small width. We do not have enough time to think over this problem, but we have some vague ideas. Based on our convergence analysis in deep linear networks, we know that the sharp rate of convergence depends on the trajectory limit. When the minimum width is sufficiently large, the trajectory limit and the initialization are not far away from each other. For deep linear networks with small widths, the result (Ziyin et al., 2022)  might shed light on convergence analysis, because the exact global minimizer could be described for a deep linear network (probably with $L_2$ regularization).
>
>
> Q6: does the main result change in a mini batch setting? the authors should suggest some answer
>
> You ask another interesting question. The convergence result of stochastic gradient descent (SGD) for deep linear residual networks (ResNet) was obtained by  Zou et al. (2020), under the condition that the initial loss should be close to the global minimum $L(W_*)$. For a large width deep linear network, our conjecture is that the SGD trajectories are also close to SGD trajectories for another corresponding convex problem around random balance initialization.
>
> Q7: the font size in the figures is too small. The authors need to update it to match the font size of the main text
>
> Thanks for your suggestion on the figures. We are supposed to do a better job on it. We update the figures to match the font size of the main text. Due to page limit, we move the figures to appendix H.
>
>
>
> REFERENCES
>
> Kawaguchi, K. (2016). Deep learning without poor local minima. Advances in neural information processing systems, 29.
>
> Ziyin, L., Li, B., & Meng, X. (2022). Exact solutions of a deep linear network. arXiv preprint arXiv:2202.04777.
>
> Zou, D., Long, P. M., & Gu, Q. (2020). On the global convergence of training deep linear ResNets. arXiv preprint arXiv:2003.01094.

---

> > ### Comment · Reviewer_gRRu · 2022-11-29
> > **reply**
> >
> > Thanks for the update. I have updated the score to recommend acceptance

---

### Official Review · Reviewer_utxC · 2022-10-23

**Confidence:** 4
**Correctness:** 4
**Technical Novelty And Significance:** 2
**Empirical Novelty And Significance:** Not applicable
**Recommendation:** 5

**Clarity, Quality, Novelty And Reproducibility:**

The paper is well-written, but I have concerns about the novelty as described above.

**Strength And Weaknesses:**

I think this paper provides a nice analysis of gradient descent on linear networks with general strongly convex (potentially over a subspace determined by data) and smooth objectives. However, I have major concerns regarding the novelty. The prior work (Du and Hu, 2019) considered gradient descent on linear networks with the squared loss, and proved linear convergence if the width is at least some constant times the depth. Compared with (Du and Hu, 2019), it seems the main innovation of this paper is an extension to general strongly convex and smooth objectives; such an extension is well-known in convex settings, and it is unclear what the main challenges and novelties are. This paper also provides convergence results on trajectories, but these results also seem to follow from strong convexity.

**Summary Of The Paper:**

This paper provides a sharp analysis for gradient descent on deep linear networks with strongly convex and smooth objectives. Specifically, for various initialization schemes such as Gaussian initialization, orthogonal initialization and balanced initialization, when the width is at least some constant times the depth, it is shown that gradient descent converges linearly to the global optimum. Moreover, it is shown that the trajectories of GD on linear networks and GD on the corresponding linear model stay close.

**Summary Of The Review:**

This paper proves linear convergence results of gradient descent on linear networks with strongly convex and smooth objectives.

---

> ### Author Response · Authors · 2022-11-19
> **Clarifications regarding novelty**
>
> We thank the reviewer for suggesting additional insightful experiments. We provide our responses below.
> For the rebuttal revision, we add a new section 4 to discuss the insights of our main theorems as well as combine the original theorem 3.1 and theorem 3.2 into new theorem 3.1.
>
> Q1: it seems the main innovation of this paper is an extension to general strongly convex and smooth objectives; such an extension is well-known in convex settings
>
> While one of the main results (Thm 3.1) and the proof technique are similar to that of Du and Hu (2019), we would like to stress that our main Theorem 3.2 is novel. To the best of knowledge, this is the first paper that reveals that the trajectory of the overparameterized deep linear neural networks is close to the original convex problem with an appropriately rescaled learning rate.
>
> Furthermore, in our new section 4, we discuss some high-level insights on GD to train deep linear networks. In this section, we answer the following questions:
>
> 1.Why GD trajectories for overparameterized deep linear neural networks with approximate balanced initialization are close to those for convex problems?
>
> 2.Why do bad saddles not affect GD for overparameterized deep linear neural networks?
>
> 3.Why does the width have to be large?
>
> The first question is related to the insights of Theorem 3.2. The key observation is that the discrete dynamics for product matrices is given by
> $$a_NW_{N:1}(t+1)=a_NW_{N:1}(t)-\eta \cdot  P(t)[\nabla L(a_NW_{N:1}(t)P_X)]+a_NE(t),$$
> where we novelty proved that the linear operator $P(t)\[\cdot\]\approx \frac{N}{n_N}\times$ identity operator. Based on this, we show that (see Lemma D3)
> $$a_NW_{N:1}(t+1)=a_NW_{N:1}(t)-\frac{N}{n_N}\eta \nabla L(a_NW_{N:1}(t)) +R(t).$$
> Without the $R(t)$ term, the discrete dynamics is exactly the GD for a convex function. To control the distance between the trajectories of deep linear networks and the convex problem, we introduce Lemma D4 and D5.
>
> The second question is related to the existence of bad saddles for the loss surface of deep linear networks. In general, vanishing Hessian will hinder optimization. To the best of knowledge,  this is the first paper that discusses how the overparametrization around balanced initialization avoids bad saddles through the whole training process. The main results in Kawaguchi (2016) imply that for $L_2$ loss, the bad saddles always satisfy that $W_{N-1:2}$ is not full rank; and through the GD training, we can show that $\inf_{t}\sigma_{min}(W_{N-1:2}(t))>0$. A deeper reasoning is related to section 4: Why does the width have to be large?
>
> The third question is related to our novel result on overparameterization phenomena in deep linear networks. We can show that there always exists at least one global minimizer in the $(\frac{1}{N})$ neighborhood of any special balanced initialization $(W_1(0),W_2(0),\cdots,W_N(0))$, and there is  bad saddles around in this neighborhood.
>
> Q2: it is unclear what the main challenges
>
> While the statements and proofs of the Theorems B.1 and Theorem B.2 for our results are similar to that of Du and Hu (2019), we actually make the following technical extension:
>
> Compared with the main results in Du and Hu (2019), we consider hidden layers with general widths as well as general strongly convex loss instead of $l_2$ loss. Technically speaking for $l_2$ loss, without loss of generality assume $X$ to be full rank and $L(W_*)=0$, due to the decomposition method in claim B.1 from Du and Hu (2019). However, when considering general strongly convex loss, we have to directly confront low rank $X$ in our analysis. Thus, the $||\cdot||_X$ appears naturally and helps us to achieve the sharp rate of convergence in our main theorems.
>
> Compared with the techniques in Du and Hu (2019), we also use classical convex optimization techniques (such as seminorm version Polyak-Łojasiewicz inequality), as well as the classical concentration inequalities for beta distribution (such as the Chernoff type bound in Lemma F3). It is worth mentioning that the one peak random projections and embedding initiation (hidden layers with general widths) is much more complicated to analyze compared with Orthogonal initialization, since the concentration for the product of beta distribution random variables is needed.

---

### Official Review · Reviewer_GzdA · 2022-10-25

**Confidence:** 3
**Clarity, Quality, Novelty And Reproducibility:** All good.
**Correctness:** 4
**Technical Novelty And Significance:** 3
**Empirical Novelty And Significance:** 3
**Recommendation:** 8

**Strength And Weaknesses:**

Strengths:
1. This paper is well-written and easy to follow. The notations are clear.
2. This paper is well-motivated and made a solid progress in understanding the gradient decent behavior of deep linear neural networks. It shows that overparameterization leads to depth-independent convergence rate, which is not unique to orthogonal initialization, as long as the initialization falls into the convergence region.
3. The analysis seems rigorous though I didn't check the proofs thoroughly.

Weaknesses:
1. The convergence analysis is local. From the last condition of eq (29) in Appendix D, it can be seen that the initial objective function cannot be too far from the minimum value.
2. This is lack of intuitive understanding of the analysis, especially on the convergence region part.


**Summary Of The Paper:**

This paper provides a convergence analysis of deep linear neural networks with overparameterization. Compared to previous work, this paper deals with different initialization methods, more general loss functions, and varying layer width and obtains sharper convergence rate.

**Summary Of The Review:**

This paper is a solid theoretical work on the convergence of gradient descent of deep linear neural networks. I recommend acceptance.

---

> ### Author Response · Authors · 2022-11-18
> **Insights for our results and analysis**
>
> Thank you for your encouraging feedback. Please see below for our responses to each individual question.
>
> In our revised paper, eq (29) is now labeled as eq (31).
>
> Q1: The convergence analysis is local. From the last condition of eq (29) in Appendix D, it can be seen that the initial objective function cannot be too far from the minimum value.--
>
> Thanks for pointing out this concern. We did not comment clearly in the paper. In our analysis, we do not require the initial objective function to be close to the minimum value. Because eq (31) basically says that
> $$
> L(a_NW_{N:1}(0)) -L(W_*)\leq \beta B_0,
> $$
> where $B_0$ only requires it to be bounded in probability. In our paper, this $B_0$ is basically equal to $B_\delta= \left(\frac{2\cdot rank(X)}{\delta}+||W_*||_X^2\right)$ defined in Lemma 2.3. Hence, the convergence analysis is not local, but we require the minimum width of the hidden layer to be large.
>
> Q2: This is lack of intuitive understanding of the analysis, especially on the convergence region part.--
>
> We add a new section 4 in our paper to discuss the insights for our results and analysis.
>
> Due to lack of description of the exact global minimizer for the deep linear network, the evolution of each $W_j$ is hard to track. Instead, we consider the discrete dynamics for product matrices (see (41) and (42):
> $$a_NW_{N:1}(t+1)=a_NW_{N:1}(t)-\eta \cdot  P(t)[\nabla L(a_NW_{N:1}(t)P_X)]+a_NE(t),$$
> where $P(t)[\cdot]$ is a linear operator such that $P(t)[\cdot] \approx \frac{N}{n_N}I$ (see (44)), where $I$ is the identity operator. $E(t)$ is negligible, which leads to without the $R(t)$ term, the discrete dynamics is exactly the GD for a convex function. To control the distance between the two trajectories, we introduce the Lemma D3, D4, D5.
>
> To help the readers to understand the insights of our results and analysis, we answer the following questions in our section 4:
>
> Why trajectories of GD for overparameterized deep linear neural networks with approximate balanced initialization are close to trajectories for convex problems?
>
> Why do bad saddles not affect GD for overparameterized deep linear neural networks?
>
> Why does the width have to be large?

---

### Official Review · Reviewer_76Qn · 2022-10-26

**Confidence:** 3
**Correctness:** 4
**Technical Novelty And Significance:** 3
**Empirical Novelty And Significance:** Not applicable
**Recommendation:** 5

**Clarity, Quality, Novelty And Reproducibility:**

The paper is well-written and delivers the main results clearly. I found the paper quite enjoyable to read.

In terms of novelty, the observation made in Theorem 3.3 (Lemmas D.3–D.5) looks novel to me, but as noted above, the remaining elements of the proof seem to rely upon existing results.

Some minor issues:
- In Eq (4), the RHS has to have a leading $\frac{\beta}{2}$ factor?
- In Definition 2.1, $n_i I_{n_i}$ -> $n_i I_{n_{i-1}}$ and $n_{j-1} I_{n_{j-1}}$ -> $n_{j-1} I_{n_j}$?
- In the definition of special balanced initialization, there is no mention on how $V_1$ is defined? Also I thought $V_N = U_{N-1}$ should also hold here?
- In the statements of Theorem 3.1 and 3.2, it looks weird that $\delta$ does not show up anywhere in the stated bound.
- In Theorem 3.3, it would be useful to mention that $q \in (0,1)$ for the step size of interest?
- The plots in Figure 1 should better be drawn in "semilogy" style?

**Strength And Weaknesses:**

Convergence of optimization methods on training linear neural networks is an important area as it can provide valuable intuitions for understanding nonlinear networks. The paper generalizes existing results and presents the main results clearly.

The $O(\kappa \log 1/\varepsilon)$ iteration complexity to achieve $\varepsilon$-suboptimality looks indeed sharp because it matches the convergence rate of GD on the convex counterpart. Nevertheless, I should mention that the $O(\kappa \log 1/\varepsilon)$ complexity was also achieved by some previous results such as Du and Hu (2019) and Hu et al. (2020).

The next main result that the trajectory of the product $W_N(t) \cdots W_1(t)$ closely follows the trajectory of $W(t)$ as we minimize the corresponding convex function $L(W)$ is something that I haven't seen in the literature, unless I missed some existing results. This part is quite intriguing as it establishes that optimizing linear NN follows a similar trajectory as the corresponding convex problem, while the problem itself is nonconvex. This observation can deliver useful insights to the community.

While I like the observation made in Theorem 3.3, for the rest of the main results (Theorems B.1, B.2, and B.3), I got the impression that the contributions made by this paper may be somewhat limited. As pointed out above, for Gaussian and orthogonal initializations, the sharp rate $O(\kappa \log 1/\varepsilon)$ was already achieved by Du and Hu (2019) and Hu et al. (2020).

A quick perusal of the proof reveals that the paper also builds on the techniques developed by these two existing papers. Remark 5 states that for the case where all hidden layers have the same width, Theorems B.1 and B.2 recover the main results of these papers. The "convergence region" established in Lemma D.1 almost exactly follows the conditions developed in the two papers. Appendix E also seems to follow the flow of the proof in Du and Hu (2019).

These observations make me question if Theorems B.1 and B.2 are merely a technical extension of the existing results on "hidden layers having identical widths" to just "hidden layers with general widths." I know this could be a false impression as I didn't go through the proof carefully; I would be happy if the authors prove me wrong.

Anyway, this concern of novelty makes me hesitate to recommend acceptance at the moment. Importantly, the main text of this paper does not make any precise comparisons against the existing results. Can you elaborate/highlight the technical challenges that had to be overcome in extending the existing results to get Theorems B.1 and B.2?


**Summary Of The Paper:**

This paper studies the convergence rates of gradient descent (GD) on minimizing the training loss $L(W_N \cdots W_1)$ of deep linear networks, where $L(W)$ is the convex training loss function of the corresponding linear model (i.e., the loss when the product $W_N \cdots W_1$ is collapsed into a single matrix $W$). We assume that $L(W)$ is strongly convex and smooth in the subspace spanned by the data points. The paper considers the following three initialization schemes

- Gaussian initialization;
- One peak random orthogonal projections and embeddings initialization (which generalizes the orthogonal initialization proposed by Hu et al. (2020));
- Special balanced initialization (which is a special case of balanced initialization considered in Arora et al. (2018a)).

For these schemes, the paper proves that, for sufficiently wide networks,

1. GD converges linearly, and the convergence rate is of the same order as minimizing $L(W)$ as a function of $W$ (Theorems B.1, B.2, and B.3; Theorems 3.1 and 3.2 in the main text are special cases of these theorems).
2. The trajectory of the product $W_N(t) \cdots W_1(t)$ as we minimize $L(W_N \cdots W_1)$ in fact stays close to the trajectory of the $W(t)$ as we minimize $L(W)$ with an appropriately rescaled learning rate (Theorem 3.3).

**Summary Of The Review:**

I enjoyed reading the paper and I think the results presented in the paper deliver valuable insights. However, at the same time, it may be the case that some of the main theorems rely too heavily on some existing results and hence are of limited novelty. In the rebuttal, it would be very helpful if the authors could clarify the technical barriers that had to be overcome in carrying out the extensions. I would be happy to raise my score if my concerns get resolved.

---

> ### Author Response · Authors · 2022-11-18
> **Clarifications regarding novelty and technical obstacles**
>
> Thank you for enjoying reading the paper as well as valuable comments, corrections and questions. Please see below for our responses to each individual question.
>
>
>
> Q1--Theorems B.1 and B.2 are merely a technical extension of the existing results--
>
> --the technical barriers that had to be overcome in carrying out the extensions--
>
> While the statements and proofs of the Theorems B.1 and Theorem B.2 for our results are similar to that of Du and Hu (2019), we actually make the following technical extension:
>
> Compared with the main results in Du and Hu (2019), we consider hidden layers with general widths as well as general strongly convex loss instead of $l_2$ loss. Technically speaking for $l_2$ loss, without loss of generality assume $X$ to be full rank and $L(W_*)=0$, due to the decomposition method in claim B.1 from Du and Hu (2019). However, when considering general strongly convex loss, we have to directly confront low rank $X$ in our analysis. Thus, the $||\cdot||_X$ appears naturally and helps us to achieve the sharp rate of convergence in our main theorems.
>
> The seminorm  as well as the restricted operator $A|_{\mathcal{R}(X)}=AP_X$ (defined at the beginning of Appendix D) help us to obtain the sharp estimate for the eigenvalues of linear operator $ P(t)[\cdot\]$ (see (44)), thus $P(t)[\cdot\]\approx \frac{N}{n_N}\times$ identity operator, where (see (42))
>
> $$a_NW_{N:1}(t+1)=a_NW_{N:1}(t)-\eta \cdot  P(t)[\nabla L(a_NW_{N:1}(t)P_X)]+a_NE(t).$$
>
>
> Compared with the techniques in Du and Hu (2019), we also use classical convex optimization techniques (such as seminorm version Polyak-Łojasiewicz inequality), as well as the classical concentration inequalities for beta distribution (such as the Chernoff type bound in Lemma F3). It is worth mentioning that the one peak random projections and embedding initiation (hidden layers with general widths) is much more complicated to analyze compared with Orthogonal initialization, since the concentration for the product of beta distribution random variables is needed.
>
> Q2: limited novelty
>
> Our novel results Theorem 3.3 (Lemmas D.3–D.5) all based on the estimation of eigenvalues of linear operator $ P(t)[\cdot\]$. In our new section 4, we discuss some insights on GD to train deep linear networks. In this section, we answer the following questions:
>
> 2.Why do bad saddles not affect GD for overparameterized deep linear neural networks?
>
> 3.Why does the width have to be large?
>
> The second question is related to the existence of bad saddles for the loss surface of deep linear networks. In general, vanishing Hessian will hinder optimization. To the best of knowledge,  this is the first paper that discusses how the overparametrization around balanced initialization avoids bad saddles through the whole training process. The main results in Kawaguchi (2016) imply that for $L_2$ loss, the bad saddles always satisfy that $W_{N-1:2}$ is not full rank; and through the GD training, we can show that $\inf_{t}\sigma_{min}(W_{N-1:2}(t))>0$. A deeper reasoning is related to section 4: Why does the width have to be large?
>
>
> The third question is related to our novel result on overparameterization phenomena in deep linear networks. We can show that there always exists at least one global minimizer in the $(\frac{1}{N})$ neighborhood of any special balanced initialization $(W_1(0),W_2(0),\cdots,W_N(0))$, and there is  bad saddles around in this neighborhood.
>
>
>
> Q2: Also I thought $V_N=U_{N-1}$ should also hold there?
>
> Thanks for pointing out the typo and concern. The reason we consider special balanced initialization is because the minimum randomness requirement for initialization is requiring only $V_N$ to be random, and $V_N\neq U_{N-1}$.
>
> Q3: it looks weird that $\delta$ does not show up anywhere in the stated bound
>
> Thanks for carefully reading the theorems. We could did a better job on this.
> The quantities $C_1,C_2,\cdots, C_6$ defined in appendix D depend on hyperparameters $n_N, \kappa, \delta, rank(X), C_0$ and $N$. I add some discussions on such quantities both at the beginning of section 3 and our new remark 6.

---

> > ### Comment · Reviewer_76Qn · 2022-12-02
> > **Thanks for the response**
> >
> > Thank you for your response! I have also checked your updated manuscript, and it seems like the newly added Section 4 is a plus to the paper; the discussions and comparisons provide useful insights.
> >
> > That said, my main worry about the novelty of this paper is only half-resolved; as far as I understood, the claimed novelty of this submission can be summarized as
> > - extension of $L(W) = \| WX-Y\|_F^2$ to general $L(W)$, which gives rise to the need for handling low-rank $X$ directly (and semi-norm version of KL etc.);
> > - extension of orthogonal init to "one peak random orthogonal projections and embeddings initialization" which necessitates a use of concentration inequalities for beta distribution.
> >
> > While I acknowledge that the second extension in Appendix F looks novel, I am not entirely convinced if the extension in the $L(W)$ part is that much of a challenge. After the introduction of the semi-norm, the extension doesn't look that difficult to me.
> >
> > One more thing I would like to point out is: the authors note in the new Section 4 that their linear operator $P(t)$ is close to a scaled version of the identity transformation, whereas the eigenvalues of the corresponding $P_t$ in Du & Hu (2019) had dependence on $\lambda_{\max}(X^T X)$ and $\lambda_{\min}(X^T X)$. However, this does not seem to be a fair comparison and the authors should better not claim novelty based on this. In this paper's formulation, the $X$ term that introduces $\lambda_{\max}(X^T X)$ and $\lambda_{\min}(X^T X)$ in the eigenvalue bounds of Du & Hu (2019) is "hidden" inside $\nabla L(a_N W_{N:1}(t) P_X)$. Take the case $ L(W) = 0.5*\|WX-Y\|_{F}^{2} $
> > for example, which has $\nabla L(W) = (WX-Y)X^T$; the $X^T$ term is hidden in $\nabla L(a_N \cdots P_X)$ in this paper, while it is inside $P_t$ in Du & Hu (2019), giving rise to the aforementioned dependence. In conclusion, the techniques are essentially the same, modulo how we define the operator $P(t)$; I suggest the authors tone down the claim around $P(t)$.

---

### Decision · Program_Chairs · 2023-01-20

**Decision:**

Reject

**Justification For Why Not Higher Score:**

Despite the fact that it received scores straddling the accept-reject boundary, including one quite high score, this was not viewed as a borderline case because the two reviewers with the most relevant expertise recommended rejection, including the only reviewer who expressed a confidence of 4.   I (the meta-reviewer) also went over the paper in some detail, and agreed with the rejection recommendations.


**Justification For Why Not Lower Score:**

N/A


**Metareview: Summary, Strengths And Weaknesses:**

The authors obtain new bounds on the rate of convergence of gradient descent applied to deep linear networks.  The bounds hold for more general loss functions than earlier bounds.  The authors also describe general conditions on the initialization that lead to convergence, and show that some initialization schemes are likely to satisfy these conditions.

The consensus view was that this analysis was too technically incremental for publication in ICLR.  The bounds were also formulated in a way that made it difficult to understand what exactly was proved.